



# Chemical characteristics and causes of airborne particulate pollution

# in warm seasons in Wuhan, central China

X.P. Lyu[1], N. Chen[2], H. Guo[1*], L.W. Zeng[1], W.H. Zhang[3], F. Shen[2], J.H. Quan[2], N. Wang[4]

[1] Department of Civil and Environmental Engineering, The Hong Kong Polytechnic

University, Hong Kong

[2] Hubei Provincial Environment Monitoring Center, Wuhan, China

3 Department of Environmental Sciences, School of Resource and Environmental Sciences,

Wuhan University, Wuhan, China

4 Guangdong Provincial Key Laboratory of Regional Numerical Weather Prediction, Institute

of Tropical and Marine Meteorology, Guangzhou, China

*Corresponding author. Tel: +852 3400 3962; Fax: +852 2334 6389; Email:

ceguohai@polyu.edu.hk

**Abstract:** Continuous measurements of airborne particles and their chemical compositions
were conducted in May, June, October and November 2014 at an urban site in Wuhan,
Central China. The results indicated that the particle concentrations stayed at a relatively high
level in Wuhan, with the averages of $135.1 \pm 4.4$ (mean$\pm$95% interval) and $118.9 \pm 3.7$ μg/m$^3$
for $PM_{10}$, and $81.2 \pm 2.6$ and $85.3 \pm 2.6$ μg/m$^3$ for $PM_{2.5}$ in summer and autumn, respectively.
Moreover, $PM_{2.5}$ frequently exceeded the National Standard Level II (*i.e.,* daily average of 75
μg/m$^3$), and six $PM_{2.5}$ episodes were captured during the sampling campaign. The
composition analysis found that secondary inorganic ions and carbonaceous aerosols
dominated the constituents of $PM_{2.5}$. It is noteworthy that potassium (K) ($2060.7 \pm 82.3$
ng/m$^3$, $47.0 \pm 2.2\%$) was the most abundant element, implying the biomass burning in/around
Wuhan. During the episodes, carbonaceous aerosols increased significantly, a signature of
combustion activities as well. The source apportionment confirmed that biomass burning was
the main cause of $PM_{2.5}$ episodes except for case 2, with the contribution ranging from $48.0 \pm$
$1.0\%$ in case 5 to $70.1 \pm 0.5\%$ in case 3. Fugitive dust and oil refinery/usage were the main
contributors to $PM_{2.5}$ in case 2. In addition to biomass burning, the contribution of oil



refinery/usage also increased in case 5. Furthermore, the mass and proportion of $NO_3^-$ peaked
in case 6. It was found that the high levels of $NO_x$ and $NH_3$, and low temperature in case 6
were responsible for the increment of $NO_3^-$. We also found that SOC formation was
dominated by the aromatics and isoprene in autumn, and the contribution of aromatics
increased during the episodes.
**Keywords:** $PM_{2.5}$; $NO_3^-$; SOA; biomass burning; formation mechanism

**1. Introduction**
Airborne particulate pollution is distinguished by high levels of particle concentrations in the
atmosphere. With typical characteristics of reducing visibility and building up of particle
concentrations, airborne particulate pollution is also called "haze", which swept across the
whole China in recent years, particularly the northern, central and eastern China (Cheng et al.,
2014; Kang et al., 2013; Wang et al., 2013). Due to its detrimental effects on human health
(Anderson et al., 2012; Goldberg et al., 2001), atmospheric environment (Yang et al., 2012;
White and Roberts, 1977), acid precipitation (Zhang et al., 2007; Kerminen et al., 2001) and
climate change (Ramanathan etal., 2001; Nemesure et al., 1995), particulate pollution has
become major concerns of scientific communities and local governments. The China's
National ambient air quality standards issued in 2012 regulate the annual upper limit of $PM_{10}$
(*i.e.*, particulate matter with aerodynamic diameter less than 10 μm) and $PM_{2.5}$ (*i.e.*,
particulate matter with aerodynamic diameter less than 2.5 μm) as 70 and 35 μg/m$^3$, and 24-h
average as 150 and 75 μg/m$^3$, respectively (GB 3095-2012).
Numerous studies focused on the spatial and temporal variations of particle concentrations,
the chemical compositions and the cause analysis of haze events (Cheng et al., 2014; Cao et
al., 2012; Zheng et al., 2005; Yao et al., 2002). Generally, the particulate pollution was
severer in winter due to the additional emissions (*e.g.* coal burning) and unfavorable
dispersion conditions (Lyu et al., 2015a; Zheng et al., 2005). And northern China often
suffers heavier, longer and more frequent haze pollution than southern China (Cao et al.,
2012). The chemical analysis indicated that secondary inorganic aerosol (SIOA), *i.e.*, sulfate
($SO_4^{2-}$), nitrate ($NO_3^-$) and ammonia ($NH_4^+$), and secondary organic aerosol (SOA) dominated
the total mass of airborne particles (Zhang et al., 2014; Zhang et al., 2012). However, the





composition differed among the size-segregated particles. In general, the secondary
components were prone to be accumulated in small particles, in contrast to the phenomenon
that crustal elements were more enriched in larger particles (Zhang et al., 2013; Theodosi et
al., 2011). Indeed, the general characteristics of particles (*i.e.*, toxicity, radiative forcing,
acidity, etc.) are all tightly associated with the chemical compositions and physical sizes,
which therefore have been extensively studied in the field of aerosols. To better understand
and control airborne particulate pollution, the causes and formation mechanisms were often
investigated (Wang et al., 2014a and b; Kang et al., 2013; Oanh and Leelasakultum, 2011).
Apart from the unfavorable meteorological conditions, emission enhancement was often the
major culprit. With no doubt, industrial and vehicular emissions contributed greatly to the
particle mass through direct emission and secondary formation of particles from gaseous
precursors, such as sulfur dioxide ($SO_2$), nitrogen oxides ($NO_x$) and volatile organic
compounds (VOCs) (Guo et al., 2011a). In addition, some other sources in specific regions or
during specific time periods also remarkably built up the particle concentrations, *e.g.* coal
combustion in north China (Cao et al., 2005; Zheng et al., 2005) and biomass burning in
Southeast Asia (Deng et al., 2008; Koe et al., 2001). Furthermore, some studies explored the
possible formation mechanisms of main particle components, *i.e.,* SIOA and SOA, and
distinguished the contributions of different formation pathways. For example, Wang et al.
(2014) demonstrated that heterogeneous oxidation of $SO_2$ on aerosol surfaces was an
important supplementary pathway to particle-bound $SO_4^{2-}$ in addition to the gas phase
oxidation and reactions in cloud. On the other hand, it was reported that homogeneous and
heterogeneous reactions dominated the formation of $NO_3^-$ in daytime and nighttime,
respectively (Pathak et al., 2011; Lin et al., 2007; Seinfeld and Pandis, 1998). Furthermore,
biogenic VOCs and aromatics were proved to be the main precursors of SOA (Kanakidou et
al., 2005; Forstner et al., 1997).
Despite numerous studies, the full components of airborne particles were seldom reported
due to the cost of sampling and chemical analysis, resulting in a gap for comprehensive
understanding of chemical characteristics of particles. Additionally, although the causes of
particle episodes were often discussed in many case studies (Wang et al., 2014; Deng et al.,
2008), the contribution was rarely quantified. Furthermore, the formation mechanisms might





be distinctive in different circumstances. Therefore, an overall understanding of chemical
characteristics of airborne particles, the cause analysis of the particle episodes and formation
mechanisms of the enhanced species are of great value. In addition, it has become a regular
phenomenon in central China that haze pollution occurred frequently in warm seasons, while
the causes were not identified and the contributions were not quantified. Wuhan is the largest
megacity in central China, and has been suffering from severe particulate pollution in recent
years. Data indicated that the frequency of $PM_{2.5}$ exceeding the national standard level II (*i.e.*,
daily average of 75 $\mu g/m^3$) in Wuhan reached 55.1% in 2014 (Wuhan Environmental Bulletin,
2014). In warm seasons of 2014, the hourly maximum $PM_{2.5}$ (564 $\mu g/m^3$) was even higher
than that in winter (383 $\mu g/m^3$), as shown in Figure S1 in the supplementary material.
Moreover, as the air quality in Wuhan is strongly influenced by the surrounding cities, the
pollution level in Wuhan also reflects the status of the city clusters in central China. However,
previous studies (Lyu et al., 2015a; Cheng et al., 2014) were insufficient to fully understand
the properties of airborne particles in this region, particularly in warm seasons, not to mention
guiding the control strategies. As such, it is urgent to grasp the chemical characteristics of
airborne particles, and to explore the causes and formation mechanisms of the particle
episodes in Wuhan.
This study deeply analyzed the chemical characteristics of $PM_{2.5}$ in Wuhan from a full suite
of component measurement data, *i.e.*, $SO_4^{2-}$, $NO_3^-$, $NH_4^+$, organic carbon (OC) including
primary organic carbon (POC) and secondary organic carbon (SOC), element carbon (EC)
and metal elements. Furthermore, based on the analysis of meteorological conditions,
chemical signatures, source apportionment and fire spot distribution, the causes of the $PM_{2.5}$
episodes were identified and the contributions were quantified. Lastly, this study utilized a
photochemical box model incorporating master chemical mechanism (PBM-MCM) and
theoretical calculation to investigate the formation processes of $NO_3^-$ and SOC. It is the first
study to quantify the contribution of biomass burning to $PM_{2.5}$, and probe into the formation
mechanisms of both inorganic and organic components in $PM_{2.5}$ in central China.

**2. Methodology**
**2.1 Data collection**



The whole set of air pollutants were continuously monitored at an urban site in the largest
megacity of central China, *i.e.*, Wuhan. The measurement covered two periods, *i.e.*, May-June
in summer and October-November in autumn of 2014. The measured species included
particle-phase pollutants such as $PM_{10}$, $PM_{2.5}$ and particle-bound components and gas-phase
pollutants, *i.e.*, VOCs, $SO_2$, CO, NO, $NO_2$, $O_3$, $HNO_3$ (g), $NH_3$ (g), HCl (g) and etc. The
sampling site (30.54N, 114.37E) was set up in the Hubei Environmental Monitoring Center
Station, as shown in Figure 1, located in a mixed commercial and residential area, where
industries were seldom permitted. The instruments were housed in a room of a six-story
building (~18 m a.g.l.), adjacent to a main road with the straight-line distance of
approximately 15 m.
$PM_{10}$ and $PM_{2.5}$ were measured by a continuous ambient particulate monitor (Thermo
Fisher-1405D, USA). The water soluble ions (WSIs) in $PM_{2.5}$ and gases including $HNO_3$,
HCl and $NH_3$ were detected using the online ion chromatography monitor
(Metrohm-MARGA 1S, Switzerland), and an aerosol OC/EC online analyzer (Sunset-RT-4,
USA) using the thermal/optical analysis technique was utilized to resolve the carbonaceous
aerosols (OC and EC). In addition, the elements in $PM_{2.5}$ were measured with a customized
metal analyzer, which combined the X-ray fluorescence and beta-ray absorption detection
techniques. For the analysis of trace gases, *i.e.*, $SO_2$, CO, NO, $NO_2$ and $O_3$, a suite of
commercial analyzers developed by Thermo Environmental Instruments (TEI) Inc. were used,
which have been described in details in previous studies (Lyu et al. 2016; Geng et al., 2009).
Furthermore, a gas chromatography-flame ionization detector-mass spectrometry
(GC-FID-MS) system (TH_PKU-300) was used to resolve the real time data of ambient
VOCs. Details about the analysis techniques, resolution, detection limits and the protocol of
quality assurance/control have been provided in Lyu et al. (2016) and Wang et al. (2014).





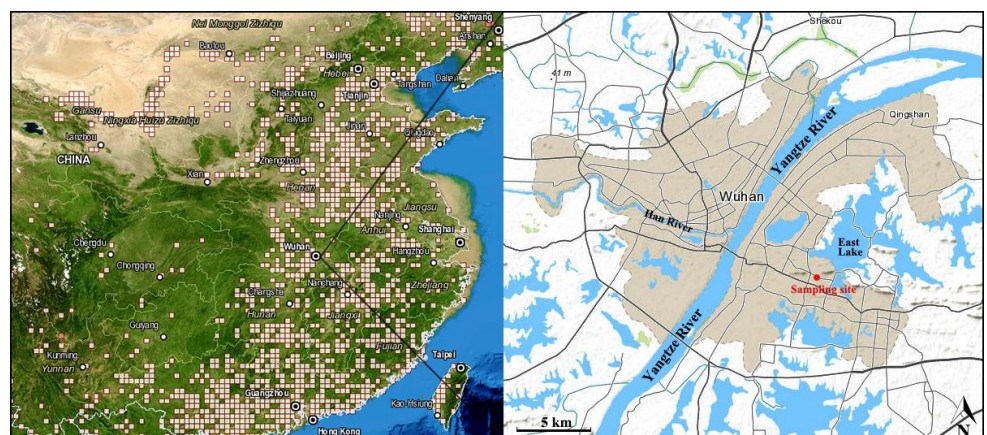

Figure 1 Geographic location of the sampling site. The white blocks in the left panel represent the

total distribution of fire spots in autumn 2014 in China, and the urban area in Wuhan is highlighted

with grey color in the right panel.

**2.2 Theoretical calculation and model simulation**

Theoretical calculation and model simulation were applied in this study to probe into the

formation mechanisms of $NO_3^-$ and SOC. The particle-bound $NO_3^-$ was generally combined

with $NH_3$ or presented as $HNO_3$ in the ammonia deficit environment, following the processes

described in R1-R3 after $HNO_3$ was formed through the oxidation of $NO_x$ (Pathak et al., 2011,

Lin et al., 2010). The production of $NO_3^-$ can be calculated by Equations 1-4.

$NH_{3\,(g)} + HNO_{3\,(g)} \leftrightarrow NH_4NO_{3\,(s)}$      $k_1 = \exp[118.87-24084/T-6.025\ln(T)]$ (ppb$^2$)      (R1)

$NH_{3\,(g)} + HNO_{3\,(g)} \leftrightarrow NH_4^+ + NO_3^-$      $k_2 = (P_1-P_2(1-a_w)+P_3(1-a_w)^2)\times(1-a_w)(1-a_w)^{1.75}k_1$ (ppb$^2$)

(R2)

$N_2O_5 + H_2O \rightarrow 2HNO_3$      $k_3 = \gamma/4(8kT/\pi m_{N2O5})0.5\,A_p$ (s$^{-1}$)      (R3)

$\ln(P_1) = -135.94 + 8763/T + 19.12\ln(T)$      (Eq.1)

$\ln(P_2) = -122.65 + 9969/T + 16.22\ln(T)$      (Eq.2)

$\ln(P_3) = -182.61 + 13875/T + 24.46\ln(T)$      (Eq.3)

$[NO_3^-] = 0.775\left(\frac{[NH_3]+[HNO_3]-\sqrt{([NH_3]+[HNO_3])^2-4([NH_3][HNO_3]-k_1(k_2))}}{2}\right)$      (Eq.4)

where R1 and R2 describe the homogeneous formation of $NO_3^-$ under the humidity of lower

and higher than deliquescence relative humidity (DRH) of $NH_4NO_3$ (*i.e.*, 62% (Tang and

Munkelwitz, 1993)), respectively. R3 presents the heterogeneous reaction of $N_2O_5$ on the

pre-existing aerosol surfaces. $k_{1-3}$ represent the rate of reactions R*1-3*. *T*, $a_w$ and *P* are the





temperature, relative humidity and the temperature-related coefficient, respectively. In R3, γ
is the reaction probability of $N_2O_5$ on aerosol surfaces, assigned as 0.05 and 0.035 on the
surface of sulfate ammonia and element carbon, respectively (Aumont et al., 1999; Hu and
Abbatt, 1997). $k$ is the Boltzmann constant ($1.38 \times 10^{-23}$), $m_{N2O5}$ is the molecular mass of
$N_2O_5$ ($1.79 \times 10^{-22}$ g), and $A_p$ is the aerosol specific surface area ($cm^2/cm^3$).
Furthermore, the PBM-MCM model was used to simulate the oxidation products in this study,
namely $O_3$, $N_2O_5$ and semi- VOCs (SVOCs), and radicals like OH, $HO_2$ and $RO_2$. With full
consideration of photochemical mechanisms and real meteorological conditions, the model
has been successfully applied in the study of photochemistry. Details about the model
construction and application can be found in Lyu et al. (2015b), Ling et al. (2014) and Lam et
al. (2013).
**2.3 Source apportionment model**
The positive matrix factorization (PMF) model was utilized to resolve the sources of $PM_{2.5}$.
As a receptor model, PMF has been extensively used in the source apportionment of airborne
particles and VOCs (Brown et al., 2007; Lee et al., 1999). Detailed introductions about the
model can be found in Paatero (1997) and Paatero and Tapper (1994). Briefly, it decomposes
the input matrix (X) into the matrices of factor contribution (G) and factor profile (F) in $p$
sources, as shown in Equation 5. A statistic value (Q) (Equation 6) was aromatically
generated to guide the selection of the best run when the lowest Q was obtained.
$x_{ij} = \sum_{k=1}^{p} g_{ik} f_{kj} + e_{ij}$       (Eq.5)
$Q = \sum_{i=1}^{n} \sum_{j=1}^{m} \left[ \dfrac{x_{ij} - \sum_{k=1}^{p} g_{ik} f_{kj}}{u_{ij}} \right]^2$       (Eq.6)
where $x_{ij}$ and $u_{ij}$ are the concentration and uncertainty of $j$ species (total of $m$) in $i$ sample
(total of $n$), $g_{ik}$ represents the contribution of $k_{th}$ source to $i$ sample, $f_{kj}$ indicates the
fraction of $j$ species in $k_{th}$ source, and $e_{ij}$ is the residual for $j$ species in $i$ sample.

**3. Results and discussion**
**3.1 Concentrations of $PM_{10}$ and $PM_{2.5}$**
Table 1 shows the mean concentrations of $PM_{10}$ and $PM_{2.5}$ in Wuhan and other Chinese cities/
regions. The average, maximum and minimum values, standard deviation or 95% confidence



interval (C.I.) were provided if available. Generally, the concentrations of airborne particles
in Wuhan ($135.1 \pm 4.4$ and $118.9 \pm 3.7$ μg/m$^3$ for PM$_{10}$; $81.2 \pm 2.6$ and $85.3 \pm 2.6$ μg/m$^3$ for PM$_{2.5}$
in summer and autumn, respectively) were lower than those in northern China (*i.e.*, Beijing
and Xi'an), comparable to those in eastern China (*i.e.*, Shanghai and Nanjing), and higher
than those in southern China (*i.e.*, Guangzhou and Hong Kong) and Taiwan. Indeed, the
sampling site, period, method and instrument all interfered with the inter-comparisons.
Bearing these factors in mind, the ambient particulate pollution in Wuhan was severe.
From summer to autumn, PM$_{10}$ experienced a considerable reduction from $135.1 \pm 4.4$ to
$118.9 \pm 3.7$ μg/m$^3$, while PM$_{2.5}$ remained statistically stable. This suggested that the emission
strength of coarse particles weakened in autumn as compared to that in summer, perhaps
related to the variations of the number of construction sites. Fugitive dust was a common
source of coarse particles, confirmed in previous studies conducted in Wuhan (Lyu et al.,
2015a; Cheng et al., 2014). Meteorological conditions were another factor for the reduction.
However, there was no significant difference in relative humidity between summer ($59.7 \pm$
$0.7\%$) and autumn ($60.4 \pm 0.8\%$) ($p > 0.05$), and the frequency of rainy days in summer
($50.8\%$) was even higher than that in autumn ($36.1\%$). Therefore, the lower PM$_{10}$ level in
autumn might benefit from the lower wind speed ($1.2 \pm 0.04$ and $0.8 \pm 0.03$ m/s in summer
and autumn, respectively; $p < 0.05$) and temperature ($25.6 \pm 0.2$ and $17.5 \pm 0.3$ ℃ in summer
and autumn, respectively; $p < 0.05$).
Table 1 Comparisons of PM$_{10}$ and PM$_{2.5}$ between Wuhan and other Chinese cities/ regions (Unit:
μg/m$^3$)

| | PM$_{10}$ | PM$_{2.5}$ | Sampling period |
|---|---|---|---|
| Wuhan | $135.1 \pm 4.4$ | $81.2 \pm 2.6$ | May-Jun. 2014 (this study) |
| | $118.9 \pm 3.7$ | $85.3 \pm 2.6$ | Oct.-Nov. 2014 (this study) |
| Beijing | 155.9 | 73.8 | Jun.-Aug. 2009 [a] |
| | 194.4 | 103.9 | Sept.-Nov. 2009 [a] |
| | $133.7 \pm 87.8$ | $71.5 \pm 53.6$ | 2012 whole year [b] |
| Xi'an | $257.8 \pm 194.7$ | $140.9 \pm 108.9$ | 2011 whole year [c] |
| Shanghai | 97.4-149.2 | 62.3-103.1 | Jul. 2009-Sept. 2010 [d] |





| Nanjing | 119-171 | 87-125 | Jun. 2012 [e] |
| Guangzhou | 23.4 | 19.2 | Jun.-Aug. 2010-2013 [f] |
|  | 51.0 | 41.3 | Sept.-Nov. 2010-2013 [f] |
| Hong Kong | $31.0 \pm 16.7$ | $17.7 \pm 12.9$ | Jun.-Aug. 2014 [g] |
|  | $55.8 \pm 23.6$ | $34.0 \pm 17.3$ | Sept.-Nov. 2014 [g] |
| Tai Wan | $39.5 \pm 11.6$ | $21.8 \pm 7.5$ | May-Nov. 2011 [h] |

[a] Liu et al. (2014); [b] Liu et al. (2015); [c] Wang et al. (2015); [d] Wang et al. (2013); [e] Shen et al.
(2014); [f] Deng et al. (2015); [g] HKEPD (2014); [h] Gugamsetty et al. (2012).

Figure 2 presents the daily concentrations of $PM_{10}$ and $PM_{2.5}$ during the sampling period in
Wuhan, with the National Standard Level II (daily averages of 150 and 75 μg/m$^3$ for $PM_{10}$
and $PM_{2.5}$, respectively). It was found that the concentrations of $PM_{10}$ and $PM_{2.5}$ frequently
exceeded the standard levels, indicating the significance of ambient particulate pollution in
Wuhan. Due to the fact that the chemical, optical and toxic properties tend to be more
apparent in smaller particles (Yang et al., 2012; Goldberg et al., 2001), and the $PM_{2.5}$
components were completely resolved, this work mainly focused on the study of $PM_{2.5}$.
During the sampling campaign, six $PM_{2.5}$ episodes named as case 1 – case 6 with the daily
averages of $PM_{2.5}$ exceeding 75 μg/m$^3$ were captured (Figure 2). It should be noted that to
ensure the data size of each episode, only the cases in which the daily $PM_{2.5}$ was
consecutively higher than 75 μg/m$^3$ for ≥3 days were treated as $PM_{2.5}$ episodes.
Table 2 summarizes the concentrations of $PM_{10}$ and $PM_{2.5}$, and the percentage of $PM_{2.5}$ in
$PM_{10}$, referred to $PM_{2.5}/PM_{10}$, during the summer and autumn episodes and non-episodes. It
was found that $PM_{10}$ and $PM_{2.5}$ increased significantly ($p<0.05$) during the episodes in both
summer and autumn. $PM_{2.5}/PM_{10}$ was a measure of the proportion of secondary species in
particles. Generally, higher $PM_{2.5}/PM_{10}$ indicates higher content of secondary aerosols, which
tend to be accumulated in smaller particles. Compared to those during non-episodes ($58.9 \pm$
1.5% and $65.3 \pm 1.3\%$ in summer and autumn, respectively), $PM_{2.5}/PM_{10}$ increased
remarkably on episode days except for case 2 ($45.9 \pm 2.5\%$), indicating that secondary
species were more enhanced during the episodes. However, the lowest $PM_{2.5}/PM_{10}$ in case 2



might imply a strong source of coarse particles. Indeed, significant contribution of fugitive
dust (5.0 ± 0.3 µg/m$^3$; 17.6 ±1.2%) was identified in case 2 (see details in section 3.3.3).

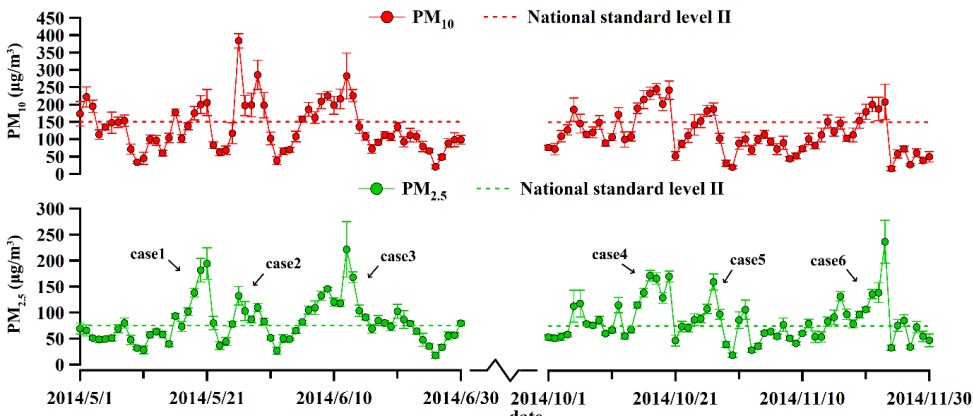


Figure 2 Daily concentrations of PM$_{10}$ and PM$_{2.5}$ in May, June, October and November 2014. Case 1:
May 16-22, case 2: May 25-30; case 3: June 5-15; case 4: October 15-20; case 5: October 24-28; Case

6: November 14-23.


Table 2 PM$_{10}$, PM$_{2.5}$ and PM$_{2.5}$/PM$_{10}$ with 95% C.I. during PM$_{2.5}$ episodes and non-episodes in Wuhan.

Non-episode 1 and Non-episode 2 represent the non-episode period in summer and autumn,

respectively.

|  | PM$_{10}$ (µg/m$^3$) | PM$_{2.5}$ (µg/m$^3$) | PM$_{2.5}$/PM$_{10}$ (%) |
|---|---|---|---|
| Case 1 | 154.3 ±10.1 | 123.0 ±9.1 | 72.8 ±2.6 |
| Case 2 | 230.1 ±19.1 | 98.9 ±5.7 | 45.9 ±2.5 |
| Case 3 | 191.4 ±9.8 | 126.7 ±7.0 | 66.9 ±1.8 |
| **Non-episode 1** | **98.5 ±3.9** | **56.6 ±1.7** | **58.9 ±1.5** |
| Case 4 | 221.8 ±8.9 | 148.6 ±5.2 | 67.9 ±2.0 |
| Case 5 | 154.2 ±10.4 | 108.2 ±6.8 | 69.3 ±3.1 |
| Case 6 | 157.3 ±9.0 | 120.0 ±7.6 | 71.2 ±2.1 |
| **Non-episode 2** | **88.7 ±3.4** | **64.2 ±2.2** | **65.3 ±1.3** |


**3.2 Chemical composition of PM$_{2.5}$**





### 3.2.1 Overall characteristics


Figure 3 shows the daily variations of $PM_{2.5}$ and its composition. Since the instrument for the
analysis of WSIs was initially deployed in September 2014, the data was not available in May
and June. The carbonaceous aerosol (18.5 ± 1.2 μg/m³) and elements (6.0 ± 0.3 μg/m³)
accounted for 19.1 ±0.6% and 6.2 ±0.2% of $PM_{2.5}$ in summer, respectively. In autumn, WSIs
was the most abundant component in $PM_{2.5}$ (64.4 ± 2.5 μg/m³; 68.6 ± 1.9%), followed by
carbonaceous aerosol (24.3 ± 1.0 μg/m³; 25.5 ± 0.8%) and elements (4.5 ± 0.2 μg/m³; 4.6 ±
0.1%). Secondary inorganic ions $SO_4^{2-}$ (18.8 ± 0.6 μg/m³), $NO_3^-$ (18.7 ± 0.8 μg/m³) and $NH_4^+$
(12.0 ± 0.4 μg/m³) dominated in WSIs, with the average contribution of 34.0 ±0.6%, 30.1 ±
0.5% and 20.4 ± 0.1%, respectively. Generally, the relative abundance of $SO_4^{2-}$ and $NO_3^-$
reflected the contribution of stationary and mobile sources to $PM_{2.5}$ (Arimoto et al., 1996).
The comparable levels of $SO_4^{2-}$ and $NO_3^-$ indicated that stationary source and mobile source
made equivalent contribution to $PM_{2.5}$ in Wuhan.

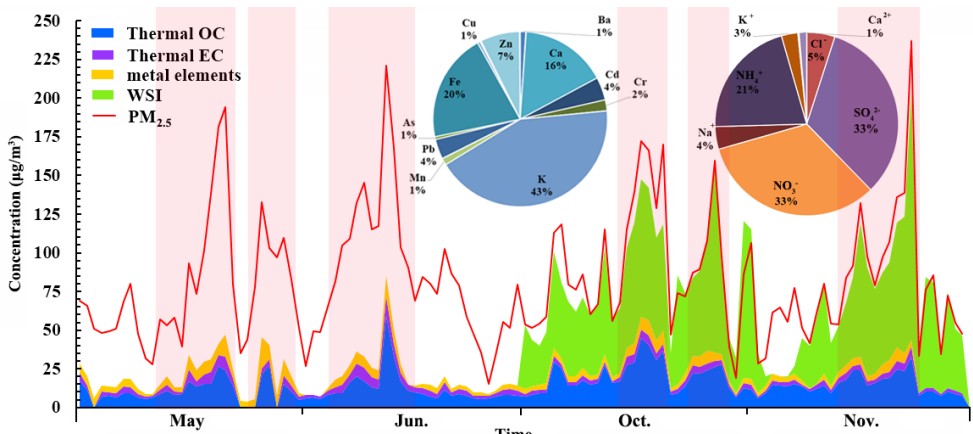

Figure 3 Daily variations of $PM_{2.5}$ and its components. The inserted pie charts represent the
composition of elements and water soluble ions, respectively. The red highlighted areas represent the
episodes.
The charge balance between the anions and cations was usually used to predict the existing
forms of the secondary inorganic ions in $PM_{2.5}$. Figure 4 shows the relative abundance of
molar charges of the anions (*i.e.*, $SO_4^{2-}$ and $NO_3^-$) and the cation (*i.e.*, $NH_4^+$). The data were
located fairly close to the one-to-one line, regardless of episode or non-episode days. This
suggested that $NH_4NO_3$ and $(NH_4)_2SO_4$ were the co-existing forms of the secondary




inorganic ions in $PM_{2.5}$ in Wuhan.

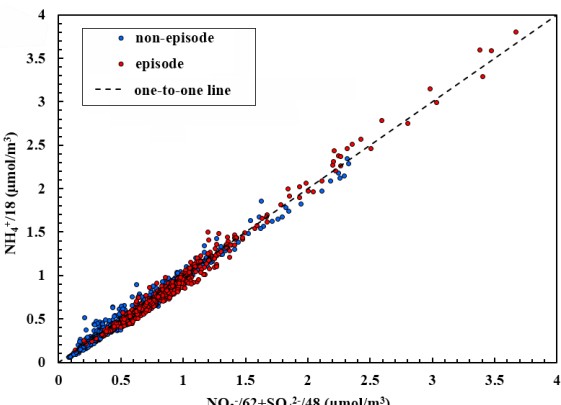


Figure 4 Charge balance between of the secondary inorganic ions in $PM_{2.5}$


For the carbonaceous aerosol, OC ($14.8 \pm 0.5$ μg/m$^3$) and EC ($3.6 \pm 0.1$ μg/m$^3$) accounted for
$79.9 \pm 0.3\%$ and $20.2 \pm 0.3\%$ of the total carbon, respectively. Generally, SOC was expected
to exist when OC/EC was larger than 2, and the proportion of SOC increased with the
increase of OC/EC ratios (Duan et al., 2005; Chow et al., 1996). The average OC/EC ratio
was $4.8 \pm 0.1$ in Wuhan, suggesting that SOC (*i.e.*, carbon fraction of SOA) was an important
component in $PM_{2.5}$. Indeed, as the constituents of OC, SOC and POC can be distinguished
with the EC-tracer method, following Equations 7 and 8 (Cabada et al., 2004):
$POC = (OC/EC)_{prim} \times EC + OC_{non\text{-}comb}$       (Eq.7)
$SOC = OC - POC$       (Eq.8)
where $(OC/EC)_{prim}$ was obtained from certain pairs of OC and EC with the OC/EC ratios
among the 10% lowest, and $OC_{non\text{-}comb}$ represented the non-combustion related OC. Figure 5
shows the linear regressions between the eligible OC and EC, where the slope and the
intercept indicated the $(OC/EC)_{prim}$ and $OC_{non\text{-}comb}$, respectively. Since the abundance of
SOC depended largely upon the oxidative capacity of the atmosphere, the oxidative radical
($HO_2$) was simulated by the PBM-MCM model and compared with the pattern of SOC. More
details about the simulation were provided in section 3.4. Figure 6 shows the hourly
concentrations of SOC and POC, and the average diurnal patterns of SOC, POC and $HO_2$. In
general, POC ($8.6 \pm 0.2$ μg/m$^3$) was slightly higher than SOC ($6.4 \pm 0.3$ μg/m$^3$) (*p* value?).





The difference reached the highest in November when the concentration was $9.5 \pm 0.4$ and
$4.7 \pm 0.3$ μg/m³ for POC and SOC, respectively. Since the production of SOC was closely
related to the atmospheric oxidative capacity, the lowest fraction of SOC in November might
be attributable to the weakest oxidative capacity, *e.g.*, $O_3$ was the lowest in November (14.3 ±
1.0 ppbv). The diurnal patterns of POC and SOC revealed that POC was relatively stable
throughout the day. The increase of POC in the early morning (06:00-08:00) and late
afternoon (16:00-20:00) was likely related to the enhanced vehicular emissions in the rush
hours, and the decrease from 08:00-15:00 might be caused by the extension of the boundary
layer. In contrast, SOC showed two peaks at ~12:00 and 19:00, which was consistent with the
diurnal variation of the simulated $HO_2$, suggesting that the formation of SOC was closely
related to the oxidative radicals in the atmosphere (detailed relationship was discussed in
section 3.4.3).

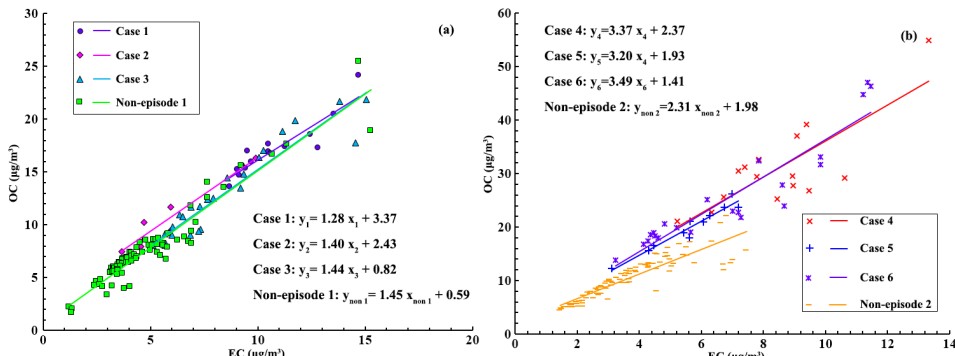


Figure 5 Regression between OC and EC with the 10% lowest OC/EC ratios in (a) summer and (b)
autumn in Wuhan

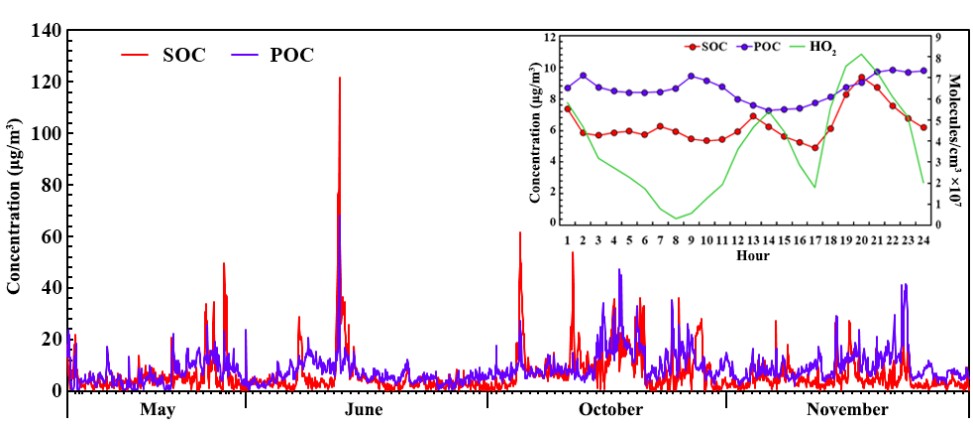




Figure 6 Hourly concentrations of SOC and POC. The insert graph presents the average diurnal
variations of SOC, POC and HO$_2$.

Among the elements, K (2060.7 $\pm$ 82.3 ng/m$^3$), Fe (996.5 $\pm$ 34.3 ng/m$^3$) and Ca (774.1 $\pm$ 39.4
ng/m$^3$) were the most abundant species, accounting for 47.0 $\pm$ 2.2%, 21.4 $\pm$ 0.3% and 15.6 $\pm$
0.3% of the total analyzed elements, respectively. Inconsistent with many other studies (Lyu
et al., 2015a; Cao et al., 2012) which reported the highest concentrations of crustal elements,
the highest K indicated the significance of biomass burning in or around Wuhan during the
monitoring period, because K is a tracer of biomass burning (Saarikoski et al., 2007; Echalar
et al., 1995).

**3.2.2 Comparison between episodes and non-episodes**
The concentrations of PM$_{2.5}$ components all increased significantly during the episodes
(Figure 3). Figure 7 presents chemical composition of PM$_{2.5}$ during episodes and
non-episodes in summer and autumn. Since WSIs data were not available in summer, the
proportions of the components were relative to the total mass resolved in PM$_{2.5}$ for cases 1-3
and non-episode 1. In summer, the fraction of OC decreased significantly ($p<0.05$) from the
non-episode 1 (57.3 $\pm$ 0.9%) to the episodes (53.0 $\pm$ 1.1%, 48.3 $\pm$ 2.6% and 55.4 $\pm$ 1.1% for
case 1, case 2 and case 3, respectively), indicating that the increment of OC was not the main
cause of the summer episodes. Conversely, the proportion of EC was significantly higher in
case 1 (19.5 $\pm$ 1.1%) (0.05$<p<$0.1, single tailed) and case 3 (22.0 $\pm$ 0.7%) ($p<0.05$, two tailed)
than that in non-episode 1 (18.7 $\pm$ 0.8%). EC was generally the tracer of incomplete
combustion (Chow et al., 1996). The accumulation of EC in PM$_{2.5}$ in case 1 and case 3
implied the enhancement of combustion source, *e.g.*, biomass burning. Furthermore, as the
indicator of biomass burning (Saarikoski et al., 2007; Echalar et al., 1995), K was remarkably
enhanced during the episodes, with the proportion in PM$_{2.5}$ increased from 8.5 $\pm$ 0.3% in
non-episode 1 to 16.1 $\pm$ 1.0% in case 1and 11.9 $\pm$ 0.3% in case 3, suggesting that biomass
burning was the leading factor of case 1 and case 3. However, consistent with that of OC, the
proportion of EC reduced to 15.0 $\pm$ 1.5% in case 2, which was just opposite to the increase of
the proportions of Ca (4.8$\pm$0.4% in non-episode 1 and 11.0 $\pm$ 1.9% in case 2) and Fe (6.3$\pm$0.3%





in non-episode 1 and 10.1 $\pm$1.2% in case 2). Since Ca and Fe were generally originated from
the crust dust, the opposite behavior of OC/EC to Ca/Fe indicated that fugitive dust rather
than combustion source led to the occurrence of case 2. It is noteworthy that the proportion of
K in case 2 (11.6 $\pm$0.7%) was also higher than that in non-episode 1 (8.5 $\pm$0.3%; $p<0.05$),
suggesting that biomass burning also made some contribution to the increment of $PM_{2.5}$ in
case 2.
In autumn, the proportion of secondary inorganic ions decreased significantly ($p<0.05$) from
non-episode 2 (70.9 $\pm$0.9%) to case 4 (62.3 $\pm$0.8%) and case 5 (67.0 $\pm$0.8%), suggesting
that the case 4 and case 5 were not directly caused by the increment of SIOA. In case 6, the
proportion of secondary inorganic ions slightly decreased to 69.3 $\pm$1.1%, while the content of
$NO_3^-$ substantially increased to 26.1 $\pm$1.0% from 19.8 $\pm$0.9% in non-episode 2. The causes
of soaring $NO_3^-$ were discussed in section 3.4.2. For the proportion of OC, it increased from
20.9 $\pm$0.8% in non-episode 2 to 27.3 $\pm$0.7% in case 4, 23.8 $\pm$1.5% in case 5 and 21.5 $\pm$0.8%
in case 6. Consistently, the concentrations of EC and K also increased during the episodes,
although there was no significance for the increments ($p>0.05$). To conclude, the
enhancement of $NO_3^-$ (only for case 6) and OC were the main causes of the autumn episodes,
which might be attributable to the biomass burning.

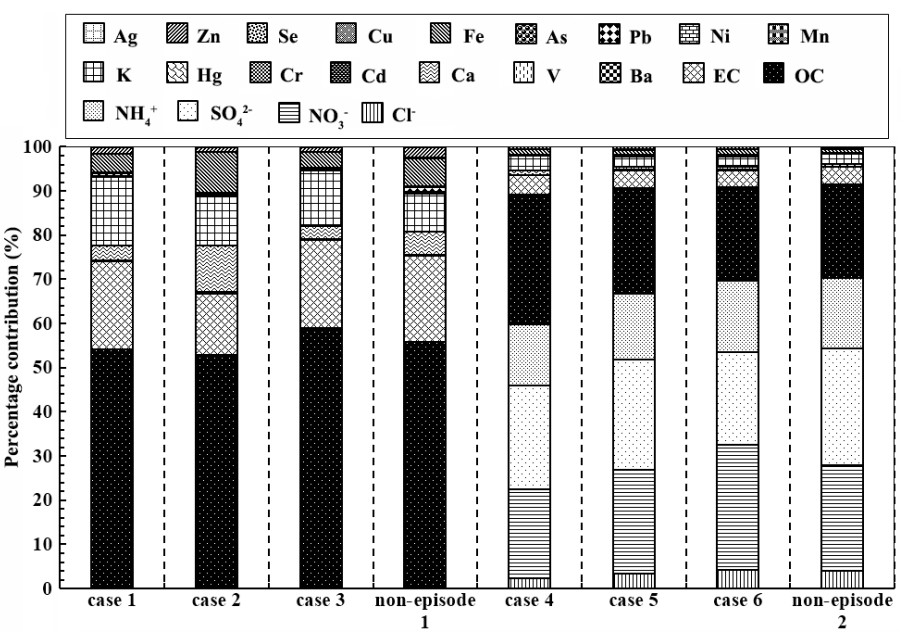






Figure 7 Chemical composition of PM$_{2.5}$ during episodes and non-episodes in summer and autumn

### 3.3 Causes of PM$_{2.5}$ episodes

### 3.3.1 Meteorological conditions

The processes of particle formation, dispersion and deposition were closely related to the
meteorological conditions. To interpret the possible causes of the PM$_{2.5}$ episodes, Figure 8
shows the patterns of wind direction/speed, temperature, relative humidity and pressure in
Wuhan during the monitoring period. Generally, the southeast winds prevailed at the
sampling site, with the wind speed of approximately 1.0 m/s. The low wind speed indicated
the dominance of local air masses. However, due to the high stability and long lifetime of
PM$_{2.5}$, the regional/superregional impact couldn't be eliminated. In comparison with those in
summer, the wind speed (summer: 1.1 $\pm$ 0.04 m/s; autumn: 0.8 $\pm$ 0.03 m/s) and temperature
(summer: 25.6 $\pm$ 0.2 m/s; autumn: 17.5 $\pm$ 0.3 m/s) were significantly ($p<0.05$) lower in
autumn, while the pressure (summer: 1006.9 $\pm$ 0.2 hPa; autumn: 1020.9 $\pm$ 0.2 hPa) was much
higher. During the episodes, the wind speed was generally lower than those in non-episodes
except for case 5. This might be one cause for the episodes, but it couldn't fully explain the
great enhancements of PM$_{2.5}$, because the wind speeds were very low and their differences
between the episodes and non-episodes were minor. Furthermore, the pressure was not very
high during the episodes except for case 6, suggesting that the synoptic system was not
responsible for the occurrence of PM$_{2.5}$ episodes.

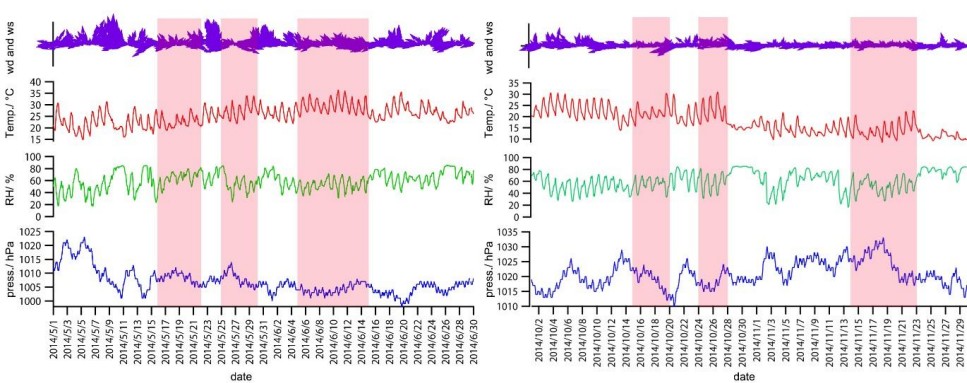

Figure 8 Meteorological patterns in Wuhan during the monitoring period. The red-highlighted areas
represent PM$_{2.5}$ episodes.



### 3.3.2 Signatures of chemical tracers

Figure 9 compares the levels of some tracer species between the episodes and non-episodes. Due to the differences in meteorological conditions and background levels, the comparisons were conducted between the episodes and non-episodes in the same season. Noticeably, the tracers of biomass burning (*i.e.*, $C_2H_2$, $CH_3Cl$, $CH_3CN$ and K) (Guo et al., 2011b; Simoneit, 2002) were much enhanced during the episodes, compared to those in non-episodes. An exception was the insignificant increment of $CH_3Cl$ in case 6, when 70% data were not available due to the instrumental error. In general, the higher levels of $C_2H_2$, $CH_3Cl$, $CH_3CN$ and K during the episodes suggested the leading role of biomass burning in building up $PM_{2.5}$. Furthermore, OC and EC also increased substantially during the episodes, consistent with the findings of previous studies (Agarwal, et al., 2010; Duan et al., 2004) that biomass burning led to great increment of the carbonaceous aerosols.

It is noteworthy that the concentrations of Ca and Fe were the most outstanding in case 2, suggesting that the fugitive dust was an important contributor to $PM_{2.5}$ in case 2, in addition to biomass burning.

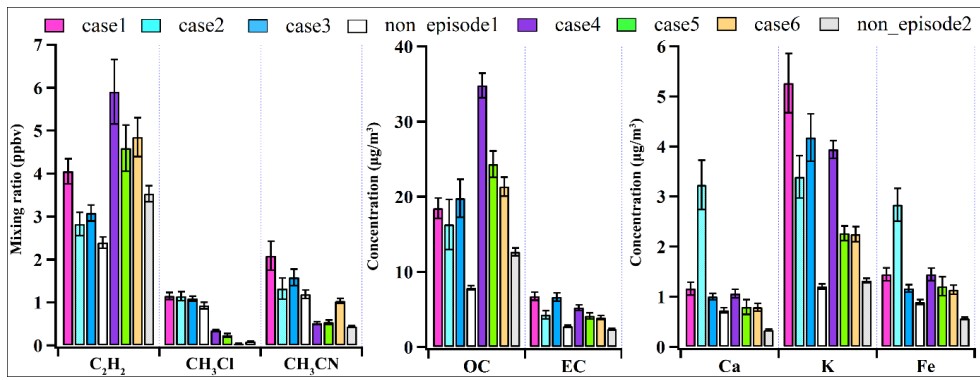

Figure 9 Concentrations of tracer species during $PM_{2.5}$ episodes and non-episodes

### 3.3.3 Source apportionment

To clarify the sources of $PM_{2.5}$ and quantify their contributions, the hourly observation data of $PM_{2.5}$ components were applied to PMF for source apportionment. Due to the lack of WSIs in May and June, the results in summer were only applicable to the sum of the resolved species in $PM_{2.5}$. Under the principle of lowest Q value, the simulations which best





reproduced the observed concentrations of all the species were selected. Figure 10 and Figure
11 present the source profiles in summer and autumn, respectively. The profiles were quite
similar in different scenarios, and four factors were resolved. Factor 1 had high loadings of
the crust elements (*i.e.*, Ba, Ca, Mn and Fe), indicating the source of fugitive dust. Factor 2
was likely associated with oil refinery and usage, in view of the high percentages of V and Ni,
which often originated from the combustion of heavy oil (Barwise et al., 1990; Nriagu and
Pacyna, 1988). Factor 3 was distinguished by the high loadings of Pb, Cu, Se and As,
generally indicating the coal combustion (Querol et al., 1995). Finally, OC, EC and secondary
inorganic ions were highly accumulated in the last factor, with the dominance of biomass
burning tracers (*i.e.*, K and Hg) (Zhang et al., 2013; Friedli et al., 2003). As such, this factor
was assigned as biomass burning.
Table 3 summarizes the mass concentrations and percentage contributions of each source to
the sum of resolved species in $PM_{2.5}$. Generally, biomass burning made the greatest
contribution to the total mass of $PM_{2.5}$ components (from $37.3 \pm 3.7\%$ in case 2 to $70.1 \pm 0.5\%$
in case 3), followed by coal combustion (from $18.5 \pm 0.9\%$ in case 3 to $44.9 \pm 1.7\%$ in
non-episode 2), oil refinery/ usage (from $7.9 \pm 0.3\%$ in case 2 to $24.4 \pm 1.4\%$ in case 5) and
fugitive dust (from $2.2 \pm 1.0\%$ in non-episode 2 to $17.6 \pm 1.2\%$ in case 2). Noticeably, in
comparison with that during non-episodes, the contributions of biomass burning were
significantly higher during the episodes, while the other sources remained relatively stable,
except for case 2 and case 5. This confirmed that biomass burning was the leading factor for
cases 1, 3, 4 and 6. In case 2, the contributions of fugitive dust ($5.0 \pm 0.3$ μg/m$^3$, $17.6 \pm 1.2\%$)
and oil refinery/usage ($6.9 \pm 0.8$ μg/m$^3$, $24.3 \pm 2.9\%$) increased considerably compared to
those in non-episode 1 ($1.2 \pm 0.1$ μg/m$^3$, $9.8 \pm 0.9\%$ and $2.4 \pm 0.1$ μg/m$^3$, $7.9 \pm 0.3\%$ for
fugitive dust and oil refinery/usage, respectively). However, the percentage contribution of
biomass burning decreased to $37.3 \pm 3.7\%$ from $41.0 \pm 0.9\%$ in non-episode 1. The variation
of source contributions suggested that fugitive dust and oil refinery/usage were the main
causes of case 2. Relative to source contributions in non-episode 2, the contribution of oil
refinery/usage in case 5 also increased by $15.1 \pm 2.6$ μg/m$^3$, in addition to the enhancement of
biomass burning ($21.8 \pm 1.7$ μg/m$^3$), revealing the combined effect of oil refinery/usage and
biomass burning on case 5.





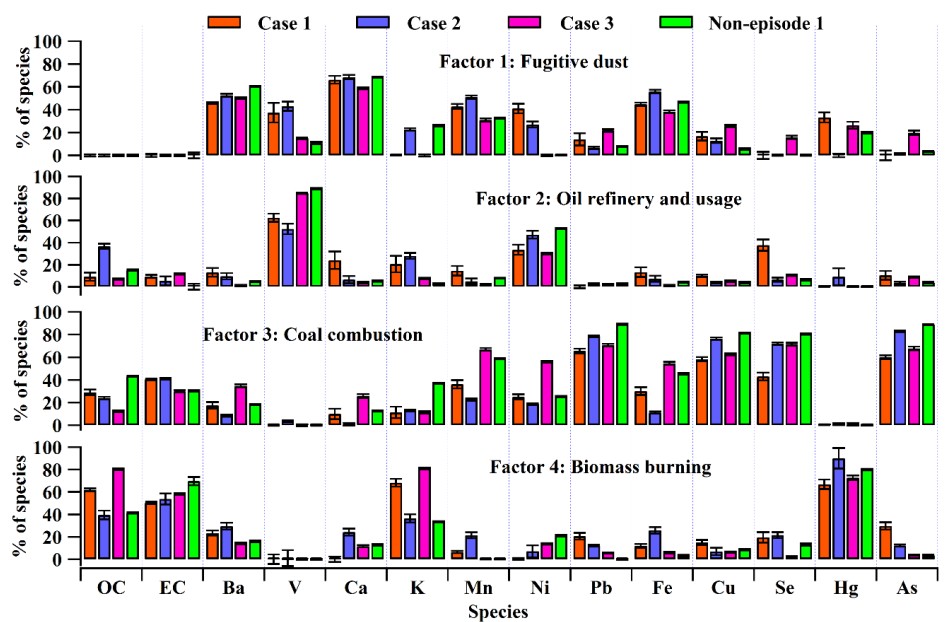


Figure 10 Profiles of the PM$_{2.5}$ sources in summer

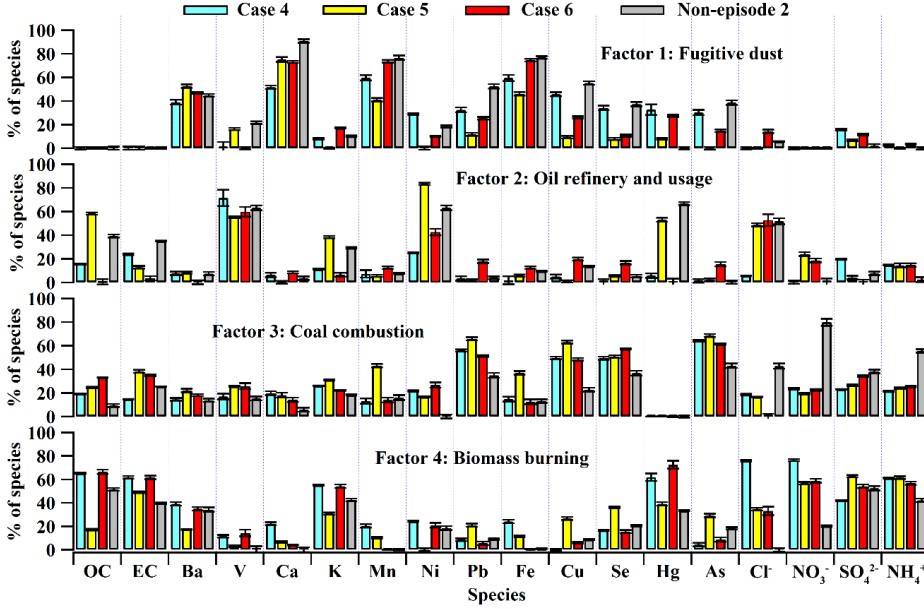



Figure 11 Profiles of the PM$_{2.5}$ sources in autumn



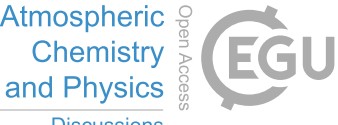


Table 3 Mass concentration ($\mu g/m^3$) and percentage contribution (shown in the bracket, %) of the
sources to the sum of the resolved species in $PM_{2.5}$

|  | Fugitive dust | Oil refinery and usage | Coal combustion | Biomass burning |
|---|---|---|---|---|
| Case 1 | 1.5 ±0.3 | 3.5 ±1.3 | 8.4 ±0.8 | **16.6 ±0.5** |
|  | (5.0 ±1.1) | (11.6 ±4.1) | (27.9 ±2.6) | **(55.4 ±1.6)** |
| Case 2 | **5.0 ±0.3** | **6.9 ±0.8** | 6.0 ±0.3 | 10.6 ±1.1 |
|  | **(17.6 ±1.2)** | **(24.3 ±2.9)** | (20.9 ±1.0) | (37.3 ±3.7) |
| Case 3 | 1.1 ±0.2 | 2.4 ±0.1 | 5.6 ±0.3 | **21.1 ±0.2** |
|  | (3.5 ±0.8) | (7.9 ±0.3) | (18.5 ±0.9) | **(70.1 ±0.5)** |
| Non-episode 1 | 1.2 ±0.1 | 1.2 ±0.1 | 4.7 ±0.03 | 4.9 ±0.1 |
|  | (9.8 ±0.9) | (9.7 ±1.0) | (39.5 ±0.2) | (41.0 ±0.9) |
| Case 4 | 6.5 ±1.1 | 14.7 ±0.8 | 24.0 ±0.5 | **67.7 ±0.7** |
|  | (5.8 ±0.9) | (13.0 ±0.7) | (21.3 ±0.4) | **(59.9 ±0.6)** |
| Case 5 | 2.8 ±0.7 | **23.1 ±1.4** | 23.3 ±0.6 | **45.4 ±0.9** |
|  | (3.0 ±0.7) | **(24.4 ±1.4)** | (24.6 ±0.7) | **(48.0 ±1.0)** |
| Case 6 | 5.2 ±0.7 | 10.0 ±2.0 | 26.2 ±0.6 | **54.8 ±1.7** |
|  | (5.4 ±0.7) | (10.4 ±2.1) | (27.2 ±0.7) | **(56.9 ±1.7)** |
| Non-episode 2 | 1.3 ±0.6 | 8.0 ±1.2 | 26.8 ±1.0 | 23.6 ±0.8 |
|  | (2.2 ±1.0) | (13.4 ±1.9) | (44.9 ±1.7) | (39.5 ±1.3) |


### 3.3.4 Open fires and air mass trajectories

To further confirm the biomass burning activities during the $PM_{2.5}$ episodes, the fire spot
distribution (downloaded from NASA Firms Web Fire Mapper, and accessible at
https://firms.modaps.eosdis.nasa.gov/firemap/) and 72-h backward air mass trajectories
(simulated by Hysplit v4.9 model) are plotted in Figure 12. Noticeably, the air masses
arriving in Wuhan had passed over the areas where the open fires were detected. In case 2,
the air mass trajectories were mainly from the south and northwest and evaded the intensive
burning areas in northeast China, which might explain why biomass burning was not a





predominant factor in case 2. In contrast, the fire spots were extremely dense in case 3 in
northeast China where the air masses originated or passed over, corresponding to the highest
contribution of biomass burning to the sum of $PM_{2.5}$ components (70.1 ±0.5%). Please note,
the fire spot distribution resolved by satellite was also influenced by meteorological
conditions. For example, the open fires were seldom observed along the air mass trajectories
in case 6, which was contradictory to the source apportionment result in which biomass
burning was the leading factor to this episode with the contribution of 56.9 ±1.7%. However,
the satellite observations revealed that the cloud cover was the highest reaching 80% in case
6 (*i.e.*, 8.0 as shown in Figure 12). This likely caused the discrepancy between the fire spot
distribution and the source apportionment. Furthermore, the temperature in case 6 (14.9 ±0.5
℃) was significantly lower than that in other cases (*p*<0.05), which might strengthen the
main causes of $PM_{2.5}$ pollution (*i.e.*, biomass burning), because the gas-to-particle conversion
was preferable at low temperature, such as the $NO_3^-$ formation discussed in section 3.4.2.

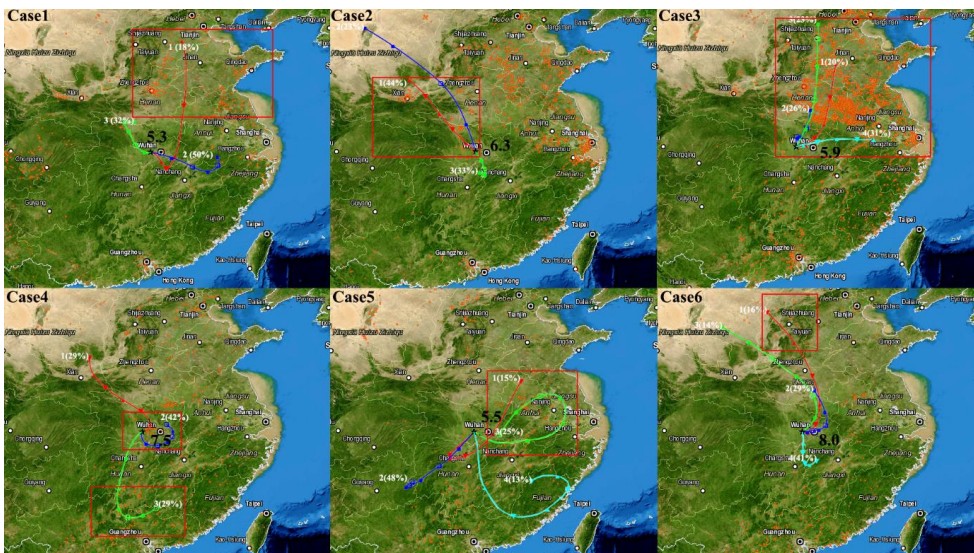

Figure 12 Fire spot distribution and 72-h backward air mass trajectories. The red squares demonstrate
the potential areas where the biomass burning aggravated particulate pollution in Wuhan. The black
figures in each case refer to the average cloud cover.
**3.4 Formation mechanisms**



### 3.4.1 Model validation


In this study, the PBM-MCM model was used to help investigate the formation mechanisms
of $NO_3^-$ and SOC. Prior to the application, the model was validated through the test of $O_3$
simulation. Figure 13 compares the daily averages and diurnal variations of $O_3$ between the
simulation and observation. It was found that the model well simulated $O_3$ variation in both
daily and diurnal patterns. However, it generally overestimated the $O_3$ levels in November.
The meteorological parameters indicated that the frequency of foggy days was extremely
high (36.7%) in November, possibly resulting in the weakening of solar radiation and
photochemical reactivity consequently. To quantitatively evaluate the performance of the
model, the index of agreement (IOA) was calculated using Equation 9.

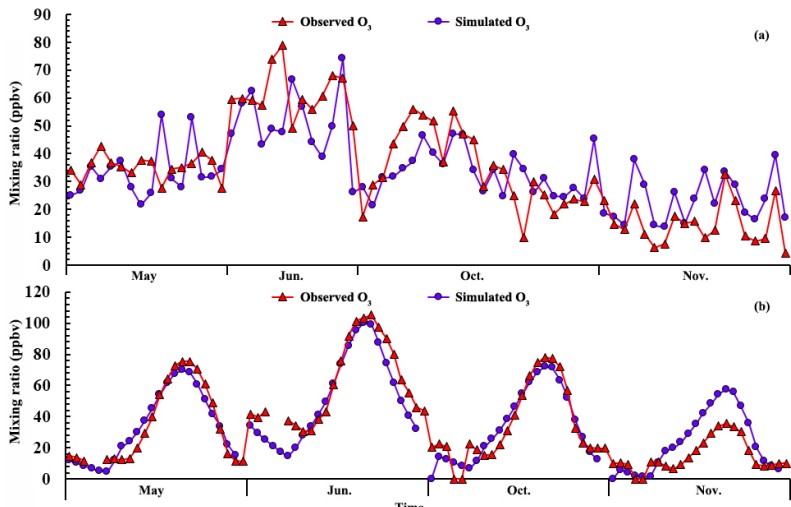


Figure 12 Comparisons of the (a) daily and (b) diurnal $O_3$ between simulation and observation. Rainy
days were excluded.

$$IOA = 1 - \frac{\sum_{i=1}^{n}(O_i - S_i)^2}{\sum_{i=1}^{n}(|O_i - \bar{O}| + |S_i - \bar{O}|)^2} \quad (Eq.9)$$

where $\bar{O}$ was the average of $n$ samples, and $O_i$ and $S_i$ represented the observed and
simulated values, respectively. Within the interval of [0, 1], Higher IOA value indicated better
agreement between the simulation and observation.
By calculation, IOA reached 0.86, indicating the excellent performance of the model in $O_3$
simulation. Since $O_3$ production is tightly associated with the oxidative radicals,



intermediates and products, the robust $O_3$ simulation gave us full confidence to accept the
simulated $N_2O_5$, $HO_2$, SVOCs, etc.

### 495    3.4.2 $NO_3^-$ formation

The composition analysis indicated that the proportion of $NO_3^-$ increased remarkably in case
6. To interpret this phenomenon, the formation mechanisms of $NO_3^-$ were investigated. Figure
14 shows the hourly variations of the calculated and observed $NO_3^-$ and the contribution of
R3 (*i.e.*, $N_2O_5 + H_2O \rightarrow 2HNO_3$), among which $NO_{3\ cal\ 1}^-$, $NO_{3\ cal\ 2}^-$ and $NO_{3\ obs}^-$ referred to
homogeneous formation (R1 and R2), total formation (R1, R2 and R3) and field
measurement of $NO_3^-$, respectively. Although the particle-bound $NO_3^-$ was influenced by
many factors (*i.e.*, formation, deposition and dispersion), the calculations generally well
reproduced the measured $NO_3^-$ in case 6, with high correlation coefficient ($R^2$ =0.63) and IOA
of 0.78. However, on November 23, 2014, the observed $NO_3^-$ decreased rapidly from 09:00,
which was not captured by the calculations. This discrepancy was likely caused by the
weather conditions on that day, because (1) the average wind speed increased from 1.7 m/s
before 09:00 to 2.7 m/s after 09:00 and even reached 4.0 m/s at 14:00; and (2) the moderate
rain began at 12:00 and continued to 23:00, with the total precipitation of 24 mm. Indeed, this
was the beginning of a 7-day rainy period, which ended case 6 with a sharp decrease of $PM_{2.5}$
concentration (approximately 175 μg/m$^3$) (see Figure 2).
As the values of $NO_{3\ cal\ 1}^-$ were very close to $NO_{3\ cal\ 2}^-$, the variation of $NO_3^-$ in case 6 could be
well explained by the homogeneous formation (R1 and R2), while the heterogeneous reaction
of $N_2O_5$ on aerosol surfaces (R3) only made minor contribution to the total $NO_3^-$ (*i.e.*, nearly
nil during 0:00-17:00, and 3.7 $\pm$ 0.6% during 18:00-23:00). Since the homogeneous
formation of $NO_3^-$ was closely related to the concentrations of $HNO_3$ (g) and $NH_3$ (g), and
temperature (see R1 and R2), Table 4 compares the temperature, $HNO_3$ (g), $NH_3$ (g), NO,
$NO_2$, $O_3$, and the simulated OH and $HO_2$ (a measure of oxidative capacity (Cheng et al.,
2010)) between case 6 and non-episode 2. It was found that $HNO_3$ (g) (0.65 $\pm$0.01 ppbv) and
$NH_3$ (g) (13.48 $\pm$ 0.72 ppbv) in case 6 were significantly higher than those during the
non-episode 2 (0.47 $\pm$ 0.03 and 9.54 $\pm$ 0.37 ppbv for $HNO_3$ and $NH_3$, respectively), which
might substantially favor the formation of $NH_4NO_3$. As $HNO_3$ (g) was generally formed



through the oxidation of $NO_x$, the production of $HNO_3$ (g) should be closely related to the
oxidative capacity of the air and the level of $NO_x$. In case 6, $O_3$ (17.09 $\pm$ 2.04 ppbv), OH ((3.8
$\pm$ 1.3) $\times 10^5$ molecules/cm$^3$) and $HO_2$ ((1.1 $\pm$ 0.3) $\times 10^7$ molecules/cm$^3$) were noticeably
lower than those in non-episode 2 ($O_3$: 24.57 $\pm$ 1.64 ppbv; OH: (7.2 $\pm$ 0.9) $\times 10^5$
molecules/cm$^3$; $HO_2$: (2.0 $\pm$ 0.2) $\times 10^7$ molecules/cm$^3$), indicating the weaker oxidative
capacity. However, NO (43.55 $\pm$ 11.65 ppbv) and $NO_2$ (44.93 $\pm$ 2.29 ppbv) were much higher
as compared to those in non-episode 2 (14.70 $\pm$ 2.40 and 29.46 $\pm$ 0.95 ppbv for NO and $NO_2$,
respectively), possibly leading to the enhancement of $HNO_3$ (g) in case 6. Furthermore, the
particle-bound $NO_3^-$ was of low thermal stability (Querol et al., 2004), and the temperature
lowered ~2.3 C$^\circ$ in case 6, which suppressed the decomposition and volatilization of
$NH_4NO_3$. Therefore, the high levels of $NO_x$ and $NH_3$, and low temperature were both
responsible for the $NO_3^-$ increment in case 6.

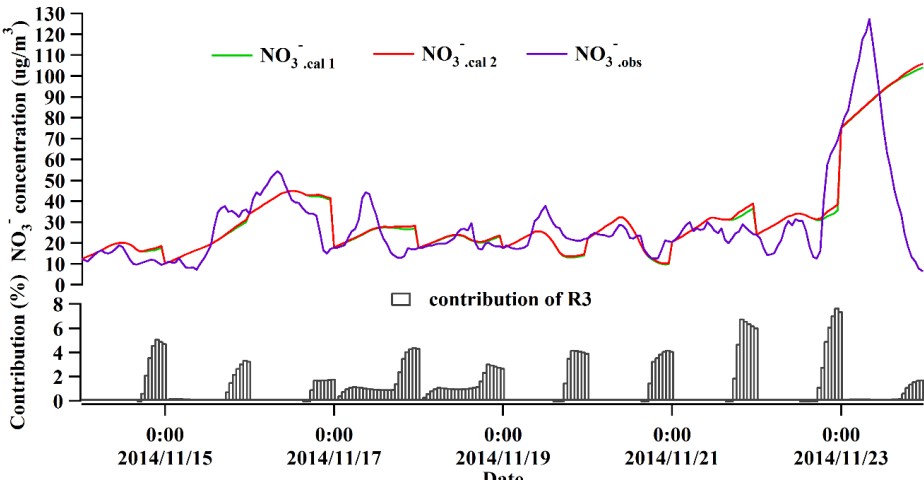

Figure 14 Comparison of $NO_3^-$ between the theoretical calculations and observation in case 6
Table 4 Comparison of temperature, $HNO_3$ (g), $NH_3$ (g), NO, $NO_2$, $O_3$ and simulated OH and $HO_2$
between case 6 and non-episode 2

|  | Case 6 | Non-episode 2 |
|---|---|---|
| Temperature (C$^\circ$) | 14.9 $\pm$ 0.5 | 17.2 $\pm$ 0.3 |
| $HNO_3$ (ppbv) | 0.65 $\pm$ 0.01 | 0.47 $\pm$ 0.03 |



| | | |
|---|---|---|
| NH$_3$ (ppbv) | 13.48 $\pm$0.72 | 9.54 $\pm$0.37 |
| NO (ppbv) | 43.55 $\pm$11.65 | 14.70 $\pm$2.40 |
| NO$_2$ (ppbv) | 44.93 $\pm$2.29 | 29.46 $\pm$0.95 |
| O$_3$ (ppbv) | 17.09 $\pm$2.04 | 24.57 $\pm$1.64 |
| OH (molecules/cm$^3$) | (3.8 $\pm$1.3) $\times 10^5$ | (7.2 $\pm$0.9) $\times 10^5$ |
| HO$_2$ (molecules/cm$^3$) | (1.1 $\pm$0.3) $\times 10^7$ | (2.0 $\pm$0.2) $\times 10^7$ |


### 3.4.3 SOC formation


Apart from high NO$_3^-$ in case 6, the proportions of OC also increased during the autumn
episodes. Since SOC is an important fraction in OC, and it often grows with the aging of air
mass, it could be of help to explain the increase of OC in autumn episodes by exploring the
possible formation mechanisms of SOC. It is well known that SOC formation is closely
related to semi-volatile oxidation products of VOCs (SVOCs), which are formed from the
reactions between oxidative radicals (*i.e.*, RO$_2$ and HO$_2$) (Kanakidou et al., 2005; Forstner et
al., 1997). Hence, the relationship between SOC and SVOCs was investigated. It should be
noted that the SVOCs were simulated by the PBM-MCM model, and SOC was calculated
with the EC-tracer method mentioned in section 3.2.1. The speciation of SVOCs and their
precursors can be found in the supplementary material Table S1. Briefly, the precursors of
SVOCs include isoprene, aromatics and C$_7$-C$_{12}$ n-alkanes.
Figure 15 presents daily and diurnal variations of SOC and SVOCs. It was found that SOC
correlated well with SVOCs in both daily (R$^2$=0.52) and diurnal (R$^2$=0.63) patterns in autumn,
indicating that the simulated SVOCs were responsible for the production of SOC. The
oxidation products of aromatics and isoprene were the main constituents of the SVOCs, with
the average contribution of 42.5 $\pm$2.8% and 39.4 $\pm$2.0%, respectively. Among the aromatics,
xylenes made the greatest contribution (15.0 $\pm$ 0.7%) to the SVOCs, followed by
trimethylbenzenes (11.5 $\pm$ 0.7%), ethylbenzene (8.8 $\pm$ 0.5%), toluene (5.1 $\pm$ 0.7%) and
benzene (2.2 $\pm$0.2%). Compared to those in non-episode 2 (*i.e.*, 40.7 $\pm$3.4% and 41.1 $\pm$2.4%
contributed by the aromatics and isoprene, respectively), the contribution of aromatics to
SVOCs increased to 46.3 $\pm$4.1% during the episodes, while the proportion of the isoprene





oxidation products decreased to $36.1 \pm 3.7\%$, suggesting that the increment of aromatics was
the main cause of the autumn episodes. To quantify the contribution of biomass burning to
SOC, the observed VOCs were apportioned to different sources, including biomass burning
with $CH_3CN$ as the tracer. The source profiles were provided in the supplementary material
Figure S2. According to the SVOCs simulated on the basis of VOCs emitted from biomass
burning, the SVOCs was elevated by $15.4 \pm 1.3\%$ due to the biomass burning during the
episodes.
In contrast, the correlations were much worse in summer ($R^2$= 0.01 and 0.31 for daily and
diurnal variations, respectively). The high frequency (50.8%) of rainy days in summer was a
factor for the poor correlation, *e.g.*, SOC was pretty low during the late period of June when
the precipitation lasted for about 10 days, while the model overestimated the SVOCs without
considering the influence of precipitation. The correlations between SOC and SVOCs ($R^2$=
0.14 and 0.19 for the daily and diurnal variations, respectively) were still poor after the rainy
days were excluded. Hence, the poor correlation should also relate to other factors such as
incomplete consideration of the contribution of biogenic VOCs. Although isoprene was
included as a precursor of the SVOCs, the other biogenic species (*i.e.*, α-pinene, β-pinene and
monoterpenes) were not monitored in this study, which were proven as important precursors
of SOC (Kanakidou et al., 2005). Moreover, the level of biogenic VOCs was much higher in
summer than that in autumn. Taking isoprene as an example, the mixing ratio of isoprene was
$66.7 \pm 4.9$ pptv in summer and only $37.2 \pm 2.6$ pptv in autumn. The higher missing level of
biogenic VOCs in summer led to a higher deficit of SVOCs, perhaps causing the poorer
correlation between SOC and SVOCs. Nevertheless, this needs further validation with more
comprehensive data of biogenic VOCs.





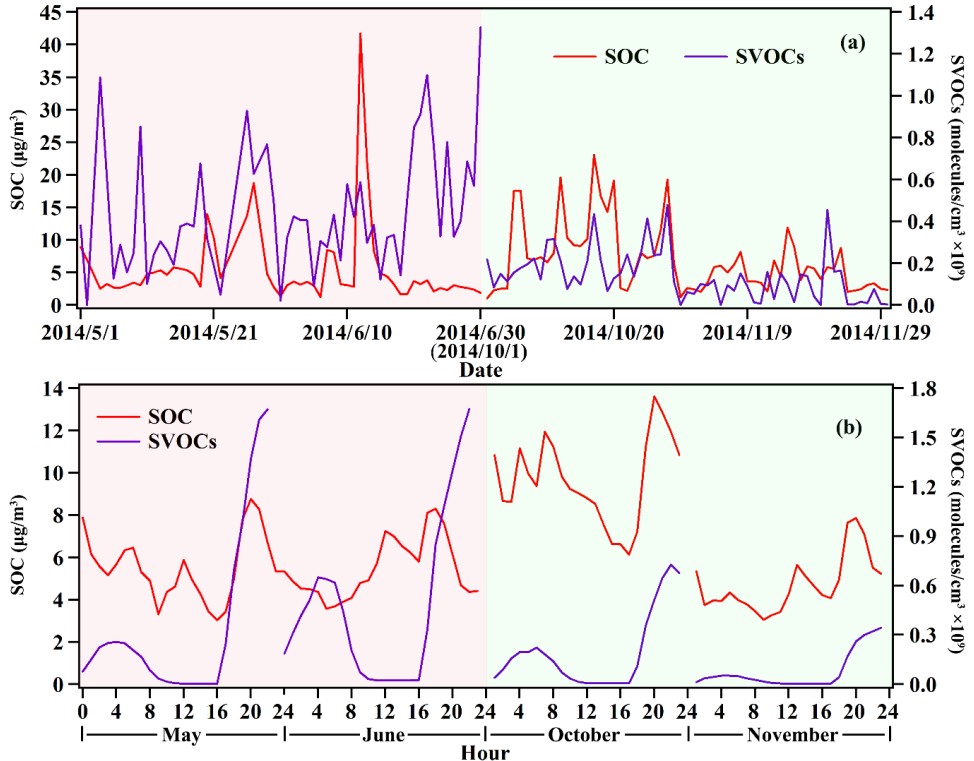

Figure 14 (a) Daily and (b) diurnal variations of SOC and SVOCs. The red and green highlighted

areas represent the summer and autumn period, respectively.

## 4. Conclusions

In summer and autumn 2014, the concentrations of $PM_{2.5}$ and its components were

continuously monitored in Wuhan, among which six $PM_{2.5}$ episodes were captured. The

analysis of $PM_{2.5}$ concentration and compositions found that Wuhan suffered from relatively

high level of $PM_{2.5}$, even in the warm seasons. Secondary inorganic ions were the most

predominant species in $PM_{2.5}$ in the form of $NH_4NO_3$ and $(NH_4)_2SO_4$. The comparable levels

of $SO_4^{2-}$ and $NO_3^-$ indicated that stationary and mobile sources were equivalently important in

Wuhan. With the EC-tracer method, it was found that POC was slightly higher than SOC, and

they both increased significantly during the episodes. K was the most abundant element,

implying the biomass burning in/around Wuhan during the sampling campaign. Indeed, the

source apportionment revealed that biomass burning was the greatest contributor to $PM_{2.5}$



during the episodes except for case 2. Fugitive dust and oil refinery/usage were the main
causes of case 2. Study of the formation mechanism of $NO_3^-$ and SOC found that $NO_3^-$ was
mainly generated from the homogeneous reactions in case 6, and the high levels of $NO_x$ and
$NH_3$, and the low temperature caused the increment of $NO_3^-$. Furthermore, the daily and
diurnal variations of SOC correlated well with those of SVOCs in autumn. The aromatics and
isoprene were the main precursors of SOC, and the contribution of aromatics increased
during the episodes. However, the correlation between SOC and SVOCs was much worse in
summer, possibly resulting from the incompleteness of biogenic VOCs input in simulating
the SVOCs. This study provided comprehensive knowledge on the chemical characteristics of
$PM_{2.5}$ in warm seasons in Wuhan, and for the first time quantified the contribution of biomass
burning to $PM_{2.5}$. The investigation of SOC formation will also inspire the application of the
explicit chemical mechanisms on the study of SOA.

**Acknowledgments:** This study was supported by the Research Grants Council of the Hong
Kong Special Administrative Region via grants PolyU5154/13E, PolyU152052/14E and
CRF/C5022-14G, and the Hong Kong Polytechnic University PhD scholarships (project
#RTUP). This study is partly supported by the Hong Kong PolyU internal grant (1-ZVCX
and 4-BCAV) and the National Natural Science Foundation of China (No. 41275122).

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
