# Peer review of "Chemical characteristics and causes of airborne particulate pollution in warm"

_Atmospheric Chemistry and Physics, 2016_

## Referee Comment (RC1) · Anonymous Referee #3 · 23 Jun 2016

The manuscript by Lyu et al. offers interesting results on the chemical compositions of PM and precursors in Wuhan China, in specific campaigns in 2014. In my opinion, results are noteworth but several revisions are necessary before pubblication in ACP :
- Abstract needs signicant improvement. It should be self-explanatory. For examples cases explanation is missing. K is 47% of what? etc. - Introduction. The first sense is not needed. - English should be significantly improved. Some sentences are too simple for a scientific journal. Some terms eed corrections (ammonia in page 2; the use of past tense should be avoided for general sentences like EC was the typical tracer of uncomplete combustion; aromatically instead of automatically etc....). Syntaxis also needs improvement. - I advice the use of SIA instead of SIOA - do authors mean

wildfires with the term "fire spot"? - Information on traffic volume next to the monitoring site is lacking - The TEOM model doe snot include FDMS. Authors should discuss what uncertainty does this add to the conclusions. -rephrase sentence in rows 199-200 and 223-224. Wht do you mean with "completely" in row 225? - Row 233: not fully true, also primary OC and EC are fine particles. - I would rather use the temr episode instead of case - Row 240: the contribution of fugitive dust is estimated in 5 ug/m3, well beofre the Source aportionment section. Please reorder. - Realting sulfate to point source and nitrate to mobile ones, is too simplistic. Traffi also emit primary particles, and no3 can also come form industries. - How OCnon-comb was estimated?? the reference Cabada et al., 2004 is missing in the Bivliography. More clarification is needed here. Do they mean that biogenic OC is all primary?? - Row 295: p avlue missing. - Row 280: provide rreferences for value of 2. - Figure 6: how authors interpret daily variation of HO2 - Section 3.2.2. there is a constrast between the conslusion that all PM componnets increase during episodes (rwo 234) and theta the OC decreases (row 329) - If cases 1 and 3 are attributed to Biomasss burning, why OC decreases? BB is the largest source sof OC as, shown in Figures 10 and 11. - Ca and Fe also come from traffic and construction/demolition works. And K in case 2 can also be emitted by mineral sources. - The source apportionment section lacks of many details which are needed to ensure that the solution is the omore realistic one. Why authors decide to perform separate PMFs for different cases? They at least present aldso the total (asembled) PMF, which will certainly imporve statistically siginifance and redice random and rotational errors of the solution. - The lack of SIA in summer is a critical issue. They are a major contributor to the mass, so the PM source apportiomnet has certanly larger errors thatn in autumn. THis needs to be discussed, showing residuals of PM and perofrmeing error estimate tools, such as BS, DISP and BS-DISp which are implemented in EPA PMFv5. By the way which software have ben used? What uncertainty of the data have been used as input? Was the Q -values the only criterion used for number of factor selection? What about distribution of residuals, G space plot, factor profiles in g/g (which are missing and need to be shown)....? -

Traffic is missing among the sources. This is hard to believe for a mega city. There must be a mix of sources, so that solutions with 5, 6...factors should be explored. Residuals of OC EC, Cu should be provided. - Backtrajectories are presented, but please specify wchih day did you select for each case? cases span overs several days... and the selection should be supported by some discussion. -Row 546 "between" should be replaced by "with" - The conlusion in row 608, should be revisited. The lack of SIA data in summer does not allow to draw comprehensive knowledge of PM2.5.
* * *

---

## Referee Comment (RC2) · Anonymous Referee #4 · 24 Jun 2016

The paper on chemical composition and source analysis by Lyu et al. presents interesting results for Wuhan, China and would be a good addition to the literature on the topic. The paper discusses chemical composition of PM2.5, possible sources and formation mechanisms for SIA and SOA. However, I have a few questions and suggestions for the authors before the paper is accepted for publication. Some of the sections of the paper present conflicting information, and the introduction section needs to be thoroughly edited (e.g. Lines 39-40, 46, 51-52, 60-62).

Specific Comments Lines 204-205: Please clarify- "... related to variations in number of construction sites.. " Is there a considerable variation in the number of active construction sites? It is surprising that the contribution from carbonaceous aerosols to

total PM is very low. It is somewhat counterintuitive that OC did not increase during the pollution episodes (Lines 328-338), since the increase is being attributed to biomass burning. In autumn, however, an increase in OC is associated with biomass burning. This is not coherent. Lines 268-273: This estimation can help interpret if the ions are neutralized. Consider expanding this section to include a discussion on whether or not the aerosols were neutralized during autumn season. Were the correlations between different metals analysed? Some of the K can also come from sources other than biomass burning (especially crustal material), and it would be useful to understand if that is the case. This is particularly relevant with reference to the earlier comment about construction sites. Also, was K found to be correlated with OC/EC? PMF: The authors do not provide any details about the outputs, and the process used for determining the factor number, or the stability of the factors. How was the uncertainty estimated? In the current analysis, the fourth factor (biomass burning) seems to include SIA, SOA and the typical biomass burning tracers. Is it possible to tease out secondary aerosol factors if 5-7 factor solutions are used? What was the total sample size used for PMF analysis? Is traffic not a source for PM in the sampling region? Technical comments Please proofread the manuscript carefully, and edit for language. Line 24: Contribution for PM10 or PM2.5? Methods: Were these hourly measurements? Which method was used for OC/EC analysis? Please include 1-2 lines about the custom element analyser. Table 1: For some cities (e.g. Beijing), is the reported value the arithmetic mean, or some other statistic? Please clarify. Also, change Tai Wan to Taiwan. Lines 60-62: Please rephrase Lines 194-195: Either specify when these statistical parameters are reported in Table 1, or remove this sentence. Lines 202-207: How strong was the correlation between the two fractions of PM? Was there a difference in this relationship during the episode compared to non-episode days? Lines 223-224: What does this sentence mean- "...Due to the fact that the chemical, optical and toxic properties tend to be more apparent in smaller particles.."

Lines 233-234: PM2.5 isn't entirely composed of secondary particles. Please edit the statement.

Line 295: p-value is missing Secondary inorganic aerosol is typically referred to as SIA in the literature. Consider using the same nomenclature. Section 3.4.1: What is the correlation coefficient for the observed/modelled ozone? Figure 11: For reporting PMF profiles, it would be easier if the authors split the profiles by episode-non-episode. The current plots are difficult to interpret since there is a lot of information on the same plot.

---

## Author Comment (AC1) · 28 Jul 2016

Please refer to the attached "Responses to the referees" for detail.

Please also note the supplement to this comment:
http://www.atmos-chem-phys-discuss.net/acp-2016-17/acp-2016-17-AC1-supplement.pdf

---

## Author Comment (AC2) · 28 Jul 2016

The manuscript by Lyu et al. offers interesting results on the chemical compositions of PM and precursors in Wuhan China, in specific campaigns in 2014. In my opinion, results are noteworth but several revisions are necessary before publication in ACP.

**Reply:** We thank the reviewer for his/her positive and encouraging comments. Detailed responses to the specific comments were listed point by point below.

1. Abstract needs significant improvement. It should be self-explanatory. For examples cases explanation is missing. K is 47% of what? etc.

**Reply:** Thanks for the suggestion. The abstract was substantially revised, including the points raised by the reviewer.

For details, please refer to lines 15-33.

2. Introduction. The first sentence is not needed.

**Reply:** Accepted with thanks. In addition, we also revised the second sentence as follows.

Airborne particulate pollution, also called "haze," has swept across China in recent years, particularly over its northern, central, and eastern parts (Cheng et al., 2014; Kang et al., 2013; Wang et al., 2013).

For details, please refer to lines 37-39.

3. English should be significantly improved. Some sentences are too simple for a scientific journal. Some terms need corrections (ammonia in page 2; the use of past tense should be avoided for general sentences like EC was the typical tracer of uncomplete combustion; aromatically instead of automatically etc....). Syntaxis also needs improvement.

**Reply:** Thanks for pointing out the errors. The grammatical errors and typos pointed out by the reviewer were corrected. To thoroughly improve the quality of the manuscript, we have asked an editing company to correct grammars and syntaxes before the submission of the revised version.

4. I advice the use of SIA instead of SIOA.

**Reply:** Accepted with thanks. Replacements were made throughout the whole manuscript.

5. Do authors mean wildfires with the term "fire spot"?

**Reply:** Exactly. The use of "fire spot" was replaced with "wildfires" throughout the whole manuscript.

6. Information on traffic volume next to the monitoring site is lacking.

**Reply:** Thanks for the useful comment. More details about the sampling site including traffic volume were provided as follows.

The traffic volume of the road was around 200 vehicles per hour. However, a wall (~2 m high) and several rows of trees (7 to 8 m high) were located between the road and the sampling site.

For details, please refer to lines 123-125.

7. The TEOM model does not include FDMS. Authors should discuss what uncertainty does this add to the conclusions.

**Reply:** Thanks for the comments. In this study, a filter dynamics measurement system (FDMS) was integrated into the TEOM system to correct the measurement deviation of the TEOM system caused by mass loss of semi-volatile particulate matters. Revisions were made as follows:

$PM_{10}$ and $PM_{2.5}$ were measured with a continuous ambient particulate monitor (Thermo Fisher-1405D, USA) integrated with a filter dynamics measurement system to minimize the loss of semivolatile particulate matter.

For details, please refer to lines 126-128.

8. Rephrase sentence in rows 199-200 and 223-224.

**Reply:** Accepted with thanks. The sentences were rephrased as follows.

Bearing in mind that the sampling site, period, method, and instrument all interfere with comparisons, the ambient particulate pollution in Wuhan was severe.

For details, please refer to lines 222-224.

Because smaller particles tend to pose more harm to human health and to the atmosphere due to their larger specific surface areas (Yang et al., 2012; Goldberg et al., 2001), and because the chemical compositions in $PM_{10}$ were not analyzed, this study focused mainly on $PM_{2.5}$.

For details, please refer to lines 243-245.

9. What do you mean with "completely" in row 225?

**Reply:** Thanks for pointing out the inappropriate use of "completely". Since this sentence was rewritten, this word was not used. Please refer to lines 243-245 for details.

10. Row 233: not fully true, also primary OC and EC are fine particles.

**Reply:** Thanks for the valuable comments. We fully agreed to the reviewer that primary OC and EC also constitute fine particles. The discussion in this part was revised as follows.

The $PM_{2.5}/PM_{10}$ value also increased remarkably on episode days compared to that on non-episode days, except for episode 2 (45.9% ±2.5%), which suggests that more secondary species and/or primary fine particles (e.g., primary OC and EC generated from combustion) were generated or released during the episodes. In contrast, the lower $PM_{2.5}/PM_{10}$ value during episode 2 might imply a strong source of coarse particles. Indeed, this inference was confirmed by the source apportionment analysis in section 3.3.3.

For details, please refer to lines 253-258.

11. I would rather use the term episode instead of case.

**Reply:** Accepted with thanks. Changes were made throughout the whole manuscript.

12. Row 240: the contribution of fugitive dust is estimated in 5 ug/m3, well before the Source apportionment section. Please reorder.

**Reply:** Thanks for the suggestion. The sentences were revised as follows.

In contrast, the lower $PM_{2.5}/PM_{10}$ value during episode 2 might imply a strong source of coarse particles. Indeed, this inference was confirmed by the source apportionment analysis in section 3.3.3.

For details, please refer to lines 256-258.

13. Relating sulfate to point source and nitrate to mobile ones, is too simplistic. Traffic also emit primary particles, and no3 can also come from industries.

**Reply:** The comment is highly appreciated. We fully agree with the reviewer. Since this short discussion was not closely related to the aims of this manuscript, it was removed.

14. How OCnon-comb was estimated?? the reference Cabada et al., 2004 is missing in the Bivliography. More clarification is needed here. Do they mean that biogenic OC is all primary??

**Reply:** Thanks for the question. OC $_{non-comb}$ means primary OC that is not related to combustion. As the intercept in the regression (Figure 5), OC $_{non-comb}$ did not vary with EC (the independent variable) which was primarily emitted from combustion activities. OC $_{non-comb}$ might include primary biogenic OC, however it did not mean that biogenic OC is all primary. In fact, this method has been extensively used to estimate POC and SOC (*e.g.*, Chu et al., 2005; Saylor et al., 2006). The method to estimate OC $_{non-comb}$ was further clarified, and revisions were made in the manuscript.

where (OC/EC) $_{prim}$ was the ratio of primary OC to EC, obtained from the pairs of OC and EC with the OC/EC ratios among the 10% lowest; and OC $_{non-comb}$ was the primary OC that was not related to combustion activities. These values were determined by the slope and intercept of the linear regression between primary OC and EC, respectively (Figure 5).

For details, please refer to lines 305-308.

The missing reference was added. For details, please refer to lines 603-605.

15. Row 295: p value is missing.

**Reply:** Sorry for the carelessness. "$p<0.05$" was added.

For detail, please refer to line 313.

16. Row 280: provide references for value of 2.

**Reply:** Accepted with thanks.

For details, please refer to line 298.

17. Figure 6: how authors interpret daily variation of $HO_2$.

**Reply:** Excellent question. The diurnal variation of $HO_2$ presents bimodal – one at daytime hours and another at night. The daytime pattern of $HO_2$ is typically bell-shaped peaked at noon or in early afternoon, consistent with the variation of atmospheric oxidative capacity (*i.e.*, $O_3$ generally peaks at noon or in early afternoon). Solar radiation is the main driving factor of this pattern. Namely, when the radiation is the strongest at noon, the photochemical reactions are the most intensive, which results in the highest production of $O_3$ and oxidative radicals including $HO_2$. Another peak of $HO_2$ is usually observed in the evening because reactions among alkenes and $O_3$ and $NO_3$lead to the formation of $HO_2$. Since concentrations of alkenes increase rapidly from late afternoon to the evening due to the decreasing photochemical consumption, increasing emissions from vehicles at rush hours in the evening, and reduction of boundary layer, the $HO_2$ production subsequently increases. With the consumption of $O_3$ and $NO_3$, the reactions with alkenes weaken in late evening. In addition, reactions among radicals consume $HO_2$. Hence, $HO_2$ decreases from late evening to early morning.

Overall, the mechanisms of $HO_2$ production and loss are complicated, and it is difficult to comprehensively discuss them in this paper. Therefore, we briefly explained the diurnal pattern of $HO_2$ in the revised manuscript.

Two peaks were found for the simulated diurnal pattern of $HO_2$, which might be caused by strong solar radiation at noon and in the early afternoon and by reactions among alkenes and $O_3$ and $NO_3$ at night (Emmerson et al., 2005; Kanaya et al., 1999).

For details, please refer to lines 318-320.

18. Section 3.2.2. there is a constrast between the conslusion that all PM componnets increase during episodes (rwo 324) and theta the OC decreases (row 329).

**Reply:** Sorry for the confusion. They results are not contradictory because the absolute concentrations of $PM_{2.5}$ components increased while the fraction (percentage) of OC in $PM_{2.5}$ decreased during the episodes.

To avoid confusion, both concentration and percentage were provided in Table 3 in the revised manuscript. Subsequently, the discussion was revised as follows.

Table 3 summarizes the mass concentrations and percentages of the main components in $PM_{2.5}$. The mass concentrations of $PM_{2.5}$ components significantly increased from non-episode days to episode days ($p < 0.05$). In contrast, the percentages of the chemical components in $PM_{2.5}$ varied by species. In summer, the fractions of EC and K in $PM_{2.5}$ experienced significant increases from non-episode 1 (EC, 4.8% $\pm 0.2$%; K, 2.0% $\pm 0.1$%) to episode 1 (EC, 5.7% $\pm 0.5$%; K, 4.4% $\pm 0.3$%) and episode 3 (EC, 5.3% $\pm 0.2$%; K, 3.0% $\pm 0.2$%). Because EC is the tracer of incomplete combustion (Chow et al., 1996) and K is the indicator of biomass burning (Saarikoski et al., 2007; Echalar et al., 1995), the higher percentages of EC and K in episodes 1 and 3 imply the outstanding contribution of biomass burning. In contrast, the fraction of OC in $PM_{2.5}$ remained stable on both episode and non-episode days ($p > 0.05$), possibly because the high temperatures in summer hindered the gas-to-particle partitioning of semivolatile organics (Takekawa et al., 2003). Furthermore, the percentages of Ca (2.9% $\pm 0.4$%) and Fe (2.7% $\pm 0.3$%) significantly increased during episode 2 ($p < 0.05$) compared to those in non-episode 1 (Ca, 1.1% $\pm 0.1$%; Fe, 1.5% $\pm 0.1$%), which shows that fugitive dust made a considerable contribution to $PM_{2.5}$ in episode 2. In addition, biomass burning might also have contributed to $PM_{2.5}$, in view of the increase in the percentage of K (non-episode 1, 2.0% $\pm 0.1$%; episode 2, 3.2% $\pm 0.2$%).

In autumn, the percentage of K significantly ($p < 0.05$) increased during episode 4 (3.1% $\pm 0.1$% vs. 2.1% $\pm 0.1$% in non-episode 2), as did that of OC (27.3% $\pm 0.7$% vs. 20.9% $\pm 0.8$% in non-episode 2), suggesting the dominant role of biomass burning in episode 4. Furthermore, the fractions of OC in episode 5 (23.8% $\pm 1.5$%) and $NO_3^-$ in episode 6 (26.1% $\pm 1.0$%) were obviously higher than those in non-episode 2 (OC, 20.9% $\pm 0.8$%; $NO_3^-$, 19.8% $\pm 0.9$%). Due to the complexity of the sources of OC and $NO_3^-$, the causes of episodes 5 and 6 are further explored in the following sections.

In summary, episodes 1, 3, and 4 were greatly affected by biomass burning. This finding was further confirmed by the significant increases in the gaseous tracers of biomass burning such as ethyne ($C_2H_2$) and methyl chloride ($CH_3Cl$) (Guo et al., 2011b; Simoneit et al., 2002) during these episodes ($p < 0.05$; see Figure S8 in the Supplement).

As this part was more related to the causes of $PM_{2.5}$ episodes, it was moved to section 3.3.2 "Chemical signatures" under section 3.3 "Causes of $PM_{2.5}$ episodes". For details, please refer to lines 378-404, Table 3 and Figure S8 in the supplement.

19. If cases 1 and 3 are attributed to Biomass burning, why OC decreases? BB is the largest source of OC, as shown in Figures 10 and 11.

**Reply:** Thanks for the question. As explained above, the absolute concentration of OC increased remarkably in episodes 1 and 3, while the percentage of OC in $PM_{2.5}$ decreased. Since the decrease in percentage was not significant ($p>0.05$), we revised the wording to "the fraction of OC in $PM_{2.5}$ remained stable ……". Furthermore, the possible reason for the stable percentage of OC was given.

In contrast, the fraction of OC in $PM_{2.5}$ remained stable on both episode and non-episode days ($p > 0.05$), possibly because the high temperatures in summer hindered the gas-to-particle partitioning of semivolatile organics (Takekawa et al., 2003).

For details, please refer to lines 386-389.

20. Ca and Fe also come from traffic and construction/demolition works. And K in case 2 can also be emitted by mineral sources.

**Reply:** We agreed with the reviewer. In this study, fugitive dust included dust from traffic, construction/demolition works, yard and bare soil. The following was added into the revised manuscript:

Because Fe and Ca are typical crustal elements, fugitive dust (e.g., dust from traffic, construction and demolition works, yards, and bare soil) was their most likely source.

For details, please refer to lines 342-343.

In addition, correlation analyses among the main elements were carried out. Weak correlations of K with Fe and Ca were found. Thus, the possibility of K emitted from mineral source was eliminated in this study. The following was added into the revised manuscript.

Correlation analysis indicated that Fe had good correlation with Ca ($R^2 = 0.66$; Figure S7 in the Supplement), whereas weak correlations of K with Fe ($R^2 = 0.14$) and Ca ($R^2 = 0.09$) were found, suggesting that Fe and Ca shared common sources that were different from the sources of K. Because Fe and Ca are typical crustal elements, fugitive dust (e.g., dust from traffic, construction and demolition works, yards, and bare soil) was their most likely source. In contrast, apart from emissions from mineral sources, K is also emitted from biomass burning. As such, K was

believed to be mainly emitted from biomass burning in this study, which is further supported by the moderate correlations of K with OC ($R^2 = 0.52$) and EC ($R^2 = 0.48$) because biomass burning also emits OC and EC (Saarikoski et al., 2007; Echalar et al., 1995).

For details, please refer to lines 338-347.

21. The source apportionment section lacks of many details which are needed to ensure that the solution is the omore realistic one. Why authors decide to perform separate PMFs for different cases? They at least present aldso the total (asembled) PMF, which will certainly imporve statistically siginifance and redice random and rotational errors of the solution.

**Reply:** The excellent comments are highly appreciated. The source apportionment part was revised substantially according to the reviewer's suggestions. The revisions included:

(1) Source apportionment was performed together with all case data collected in both episodes and non-episodes. However, since summer data of water soluble ions (WSIs) were not available, the source apportionments in summer (including episodes 1, 2, 3 and non-episode 1) and autumn (including episodes 4, 5, 6 and non-episode 2) were still separately conducted.

For details, please refer to section 3.3.3 "source apportionment".

(2) The selection criteria for best solution and the evidences (Q value, residual, G-space plot and etc.) were provided.

The selection of the factor number and the best solution depended upon the following criteria. (1) A lower Q value (Equation 6; a function to evaluate the model runs) was preferable. (2) The ratio between $Q_{robust}$ and $Q_{true}$ was lower than 1.5. In this study, the ratios were 0.8 and 0.9 for the summer and autumn data simulation, respectively. (3) Good agreement was shown between the predicted and observed $PM_{2.5}$. The slope and correlation coefficient ($R^2$) for the linear regression were 0.91 and 0.86 in summer and 0.95 and 0.98 in autumn, respectively, as shown in Figure S2 in the Supplement. The lower $R^2$ value seen during the summer might be due to the lack of WSI data. (4) The residuals were normally distributed between −3 and 3. Table S2 summarizes the percentage of samples with residuals between −3 and 3 for each species; the lowest percentages were 92.9% and 96.0% for Ni in summer and autumn, respectively. The scaled residuals for $PM_{2.5}$ are shown in Figure S3 in the Supplement. The percentage of residuals between −3 and 3

was comparable between summer (97.5%) and autumn (98.1%). Finally (5), no correlation was found between the factors, which was achieved by examining the G-space plots and controlled by the FPEAK model runs. Figures S4 and S5 in the Supplement present the G-space plots in summer and autumn, respectively. The low factor contributions and poor correlations indicated that rotational ambiguity was effectively controlled.

For details, please refer to lines 189-204.

22. The lack of SIA in summer is a critical issue. They are a major contributor to the mass, so the PM source apportiomnet has certanly larger errors thatn in autumn. THis needs to be discussed, showing residuals of PM and perofrmeing error estimate tools, such as BS, DISP and BS-DISp which are implemented in EPA PMFv5.

**Reply:** Many thanks for the excellent comment. The scaled residuals of $PM_{2.5}$ are provided in Figure S3 in the Supplement. It was found that the residuals were normally distributed between -3 and 3. However, the percentage of samples with residuals between -3 and 3 was comparable between summer (97.5%) and autumn (98.1%). In addition, bootstrap method was used to estimate the errors. According to the standard deviations estimated by bootstrap, we calculated 95% confidence intervals, which are shown as error bars in Figures 10 and 11. The error was even smaller in summer (0.6 µg/m$^3$; 0.7%) than that in autumn (2.6 µg/m$^3$; 3.2%).

The agreement between the predicted and observed $PM_{2.5}$ was also used to evaluate the simulation, as presented in Figure S2. The agreement was indeed lower in summer (slope=0.91; $R^2$=0.86) than that in autumn (slope=0.95; $R^2$=0.98). Relevant discussion was provided as follows.

(3) Good agreement was shown between the predicted and observed $PM_{2.5}$. The slope and correlation coefficient ($R^2$) for the linear regression were 0.91 and 0.86 in summer and 0.95 and 0.98 in autumn, respectively, as shown in Figure S2 in the Supplement. The lower $R^2$ value seen during the summer might be due to the lack of WSI data. (4) The residuals were normally distributed between −3 and 3. Table S2 summarizes the percentage of samples with residuals between −3 and 3 for each species; the lowest percentages were 92.9% and 96.0% for Ni in summer and autumn, respectively. The scaled residuals for $PM_{2.5}$ are shown in Figure S3 in the

Supplement. The percentage of residuals between −3 and 3 was comparable between summer (97.5%) and autumn (98.1%).

For details, please refer to lines 192-200, Table S2, Figure S2 and Figure S3 in the Supplement.

A bootstrap method was used to estimate the model errors, according to which 95% confidence intervals (CIs) were calculated. The 95% CI for $PM_{2.5}$ was 0.6 μg/m$^3$ (0.7% of predicted $PM_{2.5}$) in summer and 2.6 μg/m$^3$ (3.2% of predicted $PM_{2.5}$) in autumn.

For details, please refer to lines 205-207.

23. By the way which software have been used? What uncertainty of the data have been used as input? Was the Q -values the only criterion used for number of factor selection? What about distribution of residuals, G space plot, factor profiles in g/g (which are missing and need to be shown)....?

**Reply:** Thanks a lot for the questions about the PMF model. In this study, EPA PMF v5.0 was used to conduct the source apportionment analysis. The uncertainties were set as follows.

The uncertainties were $\sqrt{(10\% \times \text{concentration})^2 + DL^2}$ and 5/6×DL for the samples with concentrations higher and lower than DL, respectively.

For details, please refer to line 175 and lines 185-186.

The criteria used for the selection of factor number and the best solution were provided in the answer to comment 21, which involved Q-value, the ratio between Q $_{robust}$ and Q $_{true}$, agreement between the predicted and observed values, residuals and G-space plot. The factor profiles are provided in Figures 10 and 11 in the revised manuscript.

For details, please refer to lines 189-204 and Figures 10 and 11.

24. Traffic is missing among the sources. This is hard to believe for a mega city. There must be a mix of sources, so that solutions with 5, 6...factors should be explored. Residuals of OC, EC, Cu should be provided.

**Reply:** Thank you very much for the great comments and suggestions. In the revised manuscript, 5 and 6 sources including traffic emissions were identified in summer and autumn, respectively.

Table S2 in the supplement lists the percentage of samples with residuals between -3 and 3. Also, the residuals of OC, EC and Cu were provided here for the reviewer's reference.

For details, please refer to section 3.3.3 "Source apportionment".

Residuals of OC, EC and Cu in summer:

[Figure]

[Figure]

Residuals of OC, EC and Cu in autumn:

[Figure]

25. Back trajectories are presented, but please specify which day did you select for each case? Cases span overs several days... and the selection should be supported by some discussion.

**Reply:** Thanks for the suggestion. Yes, the backward trajectories were simulated during the entire period of each episode. The reason for this selection was also provided as follows.

Because the concentrations, compositions and source contributions of $PM_{2.5}$ were averaged over the entire period of each episode, the wildfire distribution and backward trajectories were also averaged for the entire period of each episode.

For details, please refer to lines 434-437.

26. Row 546 "between" should be replaced by "with".

**Reply:** Thanks for the correction. Replaced as suggested.

For details, please refer to line 519.

27. The conclusion in row 608, should be revisited. The lack of SIA data in summer does not allow to draw comprehensive knowledge of PM2.5.

**Reply:** Thanks for the suggestion. This sentence was revised as follows.

This study advances our understanding of the chemical characteristics of $PM_{2.5}$ in warm seasons in Wuhan and for the first time quantifies the contribution of biomass burning to $PM_{2.5}$.

For details, please refer to lines 581-582.

References:

Chu, S.H., 2005. Stable estimate of primary OC/EC ratios in the EC tracer method. Atmos. Environ. 39(8), 1383-1392.

Saylor, R.D., Edgerton, E.S., and Hartsell, B.E., 2006. Linear regression techniques for use in the EC tracer method of secondary organic aerosol estimation. Atmos. Environ. 40(39), 7546-7556.

Anonymous Referee #4

The paper on chemical composition and source analysis by Lyu et al. presents interesting results for Wuhan, China and would be a good addition to the literature on the topic. The paper discusses chemical composition of PM2.5, possible sources and formation mechanisms for SIA and SOA. However, I have a few questions and suggestions for the authors before the paper is accepted for publication. Some of the sections of the paper present conflicting information, and the introduction section needs to be thoroughly edited (e.g. Lines 39-40, 46, 51-52, 60-62).

**Reply:** The great comments and suggestions are highly appreciated, which helped us to improve the manuscript substantially. The introduction section was revised according to the reviewer's comments. In addition, the grammatical errors and typos in the manuscript were corrected by an editing company before the revised manuscript was submitted. Detailed responses to the specific comments were provided point by point below.

Airborne particulate pollution, also called "haze," has swept across China in recent years, particularly over its northern, central, and eastern parts (Cheng et al., 2014; Kang et al., 2013; Wang et al., 2013).

Numerous studies have been conducted in China to understand the spatiotemporal variations in particle concentrations, the chemical compositions, and the causes of haze events (Cheng et al., 2014; Cao et al., 2012; Zheng et al., 2005; Yao et al., 2002).

For details, please refer to lines 37-39 and lines 48-50.

Specific Comments:

1. Lines 204-205: Please clarify- ". . . related to variations in number of construction sites.. " Is there a considerable variation in the number of active construction sites? It is surprising that the contribution from carbonaceous aerosols to total PM is very low. It is somewhat counterintuitive that OC did not increase during the pollution episodes (Lines 328-338), since the increase is being attributed to biomass burning. In autumn, however, an increase in OC is associated with biomass burning. This is not coherent.

**Reply:** Thank you very much for the excellent comments. Since it is difficult to get the exact number of construction sites, we explained the variations of PM$_{10}$ as follows.

From summer to autumn, $PM_{10}$ levels declined considerably from $135.1 \pm 4.4$ to $118.9 \pm 3.7$ μg/m$^3$, whereas $PM_{2.5}$ remained statistically stable ($p > 0.05$). The higher summer $PM_{10}$ concentration was probably related to a higher load of fugitive dust. In Wuhan, the temperature (25.6 ℃ $\pm 0.2$ ℃) in summer was considerably higher than that (17.5 ℃ $\pm 0.3$ ℃) in autumn ($p < 0.05$), which led to lower water content in the soil and a higher tendency of dust suspension. In addition, the average wind speed in summer ($1.2 \pm 0.04$ vs. $0.8 \pm 0.03$ m/s in autumn) was also higher ($p < 0.05$), which could also have favored the generation of fugitive dust.

For details, please refer to lines 225-231.

In this study, carbonaceous aerosols constituted $23.2\pm0.5\%$ of $PM_{2.5}$, which was comparable to those (17.7% at an urban site and 17.8% at a suburban site) reported by Zhang et al. (2015) in Wuhan.

For the variations of OC, we are sorry for the confusion about absolute concentration and percentage. In this study, the absolute concentrations of OC and EC greatly increased in all the episodes. However, the percentage (or fraction) of OC in $PM_{2.5}$ remained stable in episode 1 and episode 3 compared to non-episode 1, while the percentage of OC increased in all episodes in autumn compared to non-episode 2. This discrepancy might be due to the fact that high temperature suppressed the gas-to-particle partitioning of semi-volatile organics in summer. To avoid confusion, the concentrations and percentages of main $PM_{2.5}$ components including OC were provided in Table 3 in the revised manuscript. And the explanation for the stable percentage of OC in summer episodes was provided as follows.

In contrast, the fraction of OC in $PM_{2.5}$ remained stable on both episode and non-episode days ($p > 0.05$), possibly because the high temperatures in summer hindered the gas-to-particle partitioning of semivolatile organics (Takekawa et al., 2003).

For details, please refer to lines 386-389 of section 3.3.2 "Chemical signatures" and Table 3.

2. Lines 268-273: This estimation can help interpret if the ions are neutralized. Consider expanding this section to include a discussion on whether or not the aerosols were neutralized during autumn season.

**Reply:** Accepted with thanks. The discussion was supplemented as follows.

When extending $NH_4^+$ to total cations ($NH_4^+$, $Ca^{2+}$, $Mg^{2+}$, $Na^+$, and $K^+$) and $NO_3^-$ and $SO_4^{2-}$ to total anions ($NO_3^-$, $SO_4^{2-}$, and $Cl^-$), the molar charges of the cations and anions were balanced (slope, 0.98; $R^2 = 0.98$), as shown in Figure S6 in the Supplement, indicating that $PM_{2.5}$ was neutralized during autumn in Wuhan.

For details, please refer to lines 288-291 and Figure S6.

3. Were the correlations between different metals analysed? Some of the K can also come from sources other than biomass burning (especially crustal material), and it would be useful to understand if that is the case. This is particularly relevant with reference to the earlier comment about construction sites. Also, was K found to be correlated with OC/EC?

**Reply:** Excellent comments. The correlation analyses were supplemented in the revised manuscript as follows.

Correlation analysis indicated that Fe had good correlation with Ca ($R^2 = 0.66$; Figure S7 in the Supplement), whereas weak correlations of K with Fe ($R^2 = 0.14$) and Ca ($R^2 = 0.09$) were found, suggesting that Fe and Ca shared common sources that were different from the sources of K. Because Fe and Ca are typical crustal elements, fugitive dust (e.g., dust from traffic, construction and demolition works, yards, and bare soil) was their most likely source. In contrast, apart from emissions from mineral sources, K is also emitted from biomass burning. As such, K was believed to be mainly emitted from biomass burning in this study, which is further supported by the moderate correlations of K with OC ($R^2 = 0.52$) and EC ($R^2 = 0.48$) because biomass burning also emits OC and EC (Saarikoski et al., 2007; Echalar et al., 1995).

For details, please refer to lines 338-347.

4. PMF: The authors do not provide any details about the outputs, and the process used for determining the factor number, or the stability of the factors. How was the uncertainty estimated? In the current analysis, the fourth factor (biomass burning) seems to include SIA, SOA and the typical biomass burning tracers. Is it possible to tease out secondary aerosol factors if 5-7 factor solutions are used? What was the total sample size used for PMF analysis? Is traffic not a source for PM in the sampling region?

**Reply:** Many thanks for the great comments. The same comments are also given by another reviewer. In the revised manuscript, new source apportionment simulations were performed, which separated the source of SIA and traffic. Details about the PMF outputs were also provided. Source profiles are shown in Figures 10 and 11. Source contributions are listed in Table 4. Besides, section 2.3 provided other details about the PMF inputs and outputs, *i.e.*, estimates of uncertainties, sample sizes, Q values, agreement between the predicted and observed values, the residuals and G-space plots. The model errors were estimated with the bootstrap method in PMF.

For details, please refer to lines 183-204, Figures 10 and 11, Table 4 and section 3.3.3 "Source apportionment".

Technical comments.

5. Please proofread the manuscript carefully, and edit for language. Line 24: Contribution for PM10 or PM2.5?

**Reply:** Thanks for the suggestion. To ensure the quality of the manuscript, the grammatical errors and typos in this manuscript were corrected by an editing company before the revised manuscript was submitted.

The contribution of 47.0 ± 2.2% was related to the total metal elements in $PM_{2.5}$. Since the abstract was substantially revised, the wording did not exist in the revised manuscript.

6. Methods: Were these hourly measurements? Which method was used for OC/EC analysis? Please include 1-2 lines about the custom element analyser.

**Reply:** Thanks for the questions and suggestion. Yes, PM and the chemical compositions were all hourly measurements. NIOSH thermal-optical transmission (TOT) method was used for OC/EC analysis. More detailed introduction about the customized metal analyzer was provided.

Hourly data were obtained for each species.

… and an aerosol OC/EC online analyzer (Sunset-RT-4, USA); the NIOSH thermal-optical transmission method was used to resolve the carbonaceous aerosols (OC and EC).

This instrument used a $PM_{2.5}$ impactor to collect the airborne particulate samples, which were analyzed by the β-ray in terms of mass concentrations. The filters loaded with particles were then sent to an x-ray fluorescence analysis system for quantitative analysis.

For details, please refer to lines 118-119, lines 130-132 and lines 133-135.

7. Table 1: For some cities (e.g. Beijing), is the reported value the arithmetic mean, or some other statistic? Please clarify. Also, change Tai Wan to Taiwan.

**Reply:** Thanks for the question and suggestion. Yes, the values cited from Liu et al. (2014) (Beijing) and Deng et al. (2015) (Guangzhou) were arithmetic means, which were specified in the revised manuscript. Tai Wan was revised to Taiwan.

For details, please refer to Table 1 and line 235.

8. Lines 60-62: Please rephrase.

**Reply:** Accepted with thanks. The sentence was rephrased as "In general, secondary species and mineral or sea salt components are prone to be apportioned in fine and coarse particles (Zhang et al., 2013; Theodosi et al., 2011)".

For details, please refer to lines 57-58.

9. Lines 194-195: Either specify when these statistical parameters are reported in Table 1, or remove this sentence.

**Reply:** Thanks for the suggestion. The statistical values in Table 1 were specified.

For details, please refer to Table 1 and line 235.

10. Lines 202-207: How strong was the correlation between the two fractions of PM? Was there a difference in this relationship during the episode compared to non-episode days?

**Reply:** Thanks for the great questions. Overall, $PM_{2.5}$ had fair correlation with $PM_{10}$ ($R^2$=0.59). The correlation was better in autumn ($R^2$=0.70) than that in summer ($R^2$=0.54). In addition, their correlation ($PM_{2.5}$ vs. $PM_{10}$) was comparable between episodes ($R^2$=0.44) and non-episodes ($R^2$=0.42). However, all these correlations cannot explain why $PM_{10}$ decreased in autumn while $PM_{2.5}$ remained stable. Combining your first specific comment, we revised this paragraph as follows.

From summer to autumn, $PM_{10}$ levels declined considerably from 135.1 $\pm$ 4.4 to 118.9 $\pm$ 3.7 $\mu g/m^3$, whereas $PM_{2.5}$ remained statistically stable ($p > 0.05$). The higher summer $PM_{10}$ concentration was probably related to a higher load of fugitive dust. In Wuhan, the temperature (25.6 ℃ $\pm$ 0.2 ℃) in summer was considerably higher than that (17.5 ℃ $\pm$ 0.3 ℃) in autumn ($p < 0.05$), which led to lower water content in the soil and a higher tendency of dust suspension. In addition, the average wind speed in summer (1.2 $\pm$ 0.04 vs. 0.8 $\pm$ 0.03 m/s in autumn) was also higher ($p < 0.05$), which could also have favored the generation of fugitive dust.

For details, please refer to lines 225-231.

11. Lines 223-224: What does this sentence mean- ". . . Due to the fact that the chemical, optical and toxic properties tend to be more apparent in smaller particles.."

**Reply:** Sorry for the confusion. This sentence was rephrased as follows.

Because smaller particles tend to pose more harm to human health and to the atmosphere due to their larger specific surface areas (Yang et al., 2012; Goldberg et al., 2001), and because the chemical compositions in $PM_{10}$ were not analyzed, this study focused mainly on $PM_{2.5}$.

For details, please refer to lines 243-245.

12. Lines 233-234: PM2.5 isn't entirely composed of secondary particles. Please edit the statement.

**Reply:** Thanks for the excellent comment. The statement "$PM_{2.5}/PM_{10}$ was a measure of the proportion of secondary species in particles." was deleted. Instead, the discussion was revised as follows.

The $PM_{2.5}/PM_{10}$ value also increased remarkably on episode days compared to that on non-episode days, except for episode 2 (45.9% $\pm$ 2.5%), which suggests that more secondary species and/or primary fine particles (e.g., primary OC and EC generated from combustion) were generated or released during the episodes. In contrast, the lower $PM_{2.5}/PM_{10}$ value during episode 2 might imply a strong source of coarse particles. Indeed, this inference was confirmed by the source apportionment analysis in section 3.3.3.

For details, please refer to lines 253-258.

13. Line 295: p-value is missing.

**Reply:** Sorry for the carelessness. It was added. For details, please refer to line 313.

14. Secondary inorganic aerosol is typically referred to as SIA in the literature. Consider using the same nomenclature.

**Reply:** Accepted with thanks. Revisions were made throughout the manuscript.

14. Section 3.4.1: What is the correlation coefficient for the observed/modelled ozone?

**Reply:** Thanks for the question. The slope and $R^2$ of linear regression between the observed and modelled ozone was 0.87 and 0.60, respectively. Since index of agreement (IOA) is often used to evaluate the model performance, we used IOA in the manuscript.

15. Figure 11: For reporting PMF profiles, it would be easier if the authors split the profiles by episode-non-episode. The current plots are difficult to interpret since there is a lot of information on the same plot.

**Reply:** Thanks for the suggestion. Since the data collected in episodes and non-episodes were combined to conduct source apportionment analysis in the revised manuscript, the average profiles in summer and autumn were given separately (Figures 10 and 11), which looked better.

For details, please refer to Figures 10 and 11.

References:

Zhang, F., Wang, Z.W., Cheng, H.R., Lv, X.P., Gong, W., Wang, X.M., and Zhang, G., 2015. Seasonal variations and chemical characteristics of $PM_{2.5}$ in Wuhan, central China. Sci. Total Environ. 518, 97-105.

---

## Author Response (AR2)

**Responses to Editor**

Comments to the Author:

Please be so kind to apply the minor changes suggested by the referee and submit again the revised paper. Also please give more details to the PMF analysis description as requested by the 2 referees and by the one that has revised the last version.

We thank the editor for kind handling of this paper. As suggested, very detailed descriptions about the configuration, run, results and error estimation of the PMF were provided in the last and present versions of the manuscript, including the inputs of concentrations and uncertainties, the S/N ratios, the Q values, the ratio between Q robust and Q true, the agreement between observed and predicted values, the residuals, the G-space plots and the errors estimated with bootstrap method. These criteria and tools fully ensured the reasonability of the source apportionment results. Detailed responses to the Referee #4's comments were provided item by item below.

**Responses to Anonymous Referee #4**

Methodology: Please mention that WSI data was not available for summer.

Thanks for the suggestion. Revisions were made as follows.

However, data were not available in May and June, because the instrument was initially deployed in September.

For details, please refer to lines 130-131, page5.

Lines 49, 60: Change chemical compositions to chemical composition

Accepted with thanks.

Line 101: Change deeply to comprehensively

Accepted with thanks.

Lines 103-104: Metal elements?

Thanks for the question. Since some elements were not metals, such as As and Se. "Metal elements" was changed to "elements" throughout the manuscript.

Figure 3: Does this figure refer to autumn data?

Thanks for the question. Actually, this figure shows the data in both summer and autumn. WSIs are represented with green areas, which are not shown in May and June due to the lack of data.

Lines 340-346: Does elemental K correlate with K+?

Good question. During the sampling period, $K^+$ correlated well with K with $R^2$ and slope of 0.88 and 0.80, respectively. Revisions were made as follows.

K$^+$ monitored by the online ion chromatography correlated well (R$^2$ = 0.88; slope = 0.80) with K monitored by the customized metal analyzer. To keep consistency with other elements, K rather than K$^+$ was used to do the following analyses in this study.

For details, please refer to lines 136-139, page 5.

Line 396: Cu is used as an antioxidant in brake pads. This is not clear. Please elaborate. Refer to the following paper for brake pad composition, and emissions: Grigoratos, T. and Martini, G. (2015) Brake pad particle emissions: a review. Environ Sci Pollut Res Intl, 22:2491-2504.

Thanks for the comment and suggestion of the reference.

Cu has been identified as one of the most abundant metals in both brake linings and the brake wear particles with the concentration of up to 210 mg/g in brake wear dust (Grigoratos and Martini, 2015).

For details, please refer to lines 426-428, page 19 and lines 692-693, page 32.

PMF model: S/N ratio? Were data points not confirming to the residual range (-3 to +3) removed from the analysis?

Thanks for the questions. The signal-to-noise (S/N) ratios were all greater than 1, indicating "good" signal for all the species used for source apportionment, according to the PMF 5.0 User Guide. Data points not confirming to the residual range (-3 to +3) were removed from the analysis. Revisions were made as follows.

The signal-to-noise (S/N) ratios were all greater than 1, indicating "good" signal for all the species involved in source apportionment, according to the PMF 5.0 User Guide.

For details, please refer to lines 190-192, page 7.

Data points not confirming to the residual range (-3 to +3) were removed from the analysis.

For details, please refer to lines 206-207, page 8.

Figure 10: It is a surprising that more EC is apportioned to biomass burning compared to vehicle emissions both in summer and autumn.

We highly appreciate the comment. As a tracer of combustion, EC is generally emitted from vehicle emissions, coal combustion, biomass burning and other combustion activities. Streets et al. (2001) reviewed previous studies and estimated the emission factors of EC in China from different sectors. Results indicated that the average emission factors were 0.08 and 1.1 g/kg fuel for gasoline and diesel vehicular emissions, respectively, while they were 0.90, 0.58 and 0.72 g/kg for the field combustion of wheat, rice and corn residuals, respectively. Since diesel vehicles only accounted for 11.1% of the vehicle fleet in Wuhan (the data were available at http://www.whepb.gov.cn/u/cms/whepb/201506/051551398i2j.pdf), it was reasonable that more EC was apportioned to biomass burning compared to vehicle emissions.

On the other hand, May and June in summer and October and November in autumn are the periods with intensive biomass burning in and/or around Wuhan, particularly the combustion of crop residuals in agricultural provinces in central China. Namely, the intensive biomass burning during the study period was another factor leading to higher EC emitted by biomass burning than vehicular emissions.

Revisions were made as follows.

Both biomass burning and vehicular emissions are important sources of EC. The much lower EC apportioned to vehicular emissions in this study is explained in Section 1 of the Supplement.

For details, please refer to lines 432-434, page 19.

Section 1 Explanation of higher EC apportioned to biomass burning compared to vehicular emissions.

As a tracer of combustion, EC is generally emitted from vehicular emissions, coal combustion, biomass burning and other combustion activities. Streets et al. (2001) reviewed previous studies and estimated the emission factors of EC in China from different sectors. Results indicated that the average emission factors were 0.08 and 1.1 g/kg fuel for gasoline and diesel vehicular emissions, respectively, while they were 0.90, 0.58 and 0.72 g/kg for the field combustion of wheat, rice and corn residuals, respectively. Since diesel vehicles only accounted for 11.1% of the vehicle fleet in Wuhan (the data were available at http://www.whepb.gov.cn/u /cms/whepb/201506/051551398i2j.pdf), it was reasonable that more EC was apportioned to biomass burning compared to vehicular emissions.

On the other hand, May and June in summer and October and November in autumn are the periods with intensive biomass burning in and/or around Wuhan, particularly the combustion of crop residuals in agricultural provinces in central China. Namely, the intensive biomass burning during the study period was another factor leading to higher EC emitted by biomass burning than vehicular emissions.

For details, please refer to Section 1 in the Supplement.

Table 4: Contribution of coal is very different between non-episodes 1 and 2. Why?

Many thanks for the question. The source apportionment results indicated that contribution of coal combustion in non-episode 2 was significantly lower than that in non-episode 1 ($p<0.05$). Bearing in mind the uncertainties caused by the lack of WSIs in summer, the lower contribution of coal combustion in non-episode 2 might be attributable to the National Day holiday from October 1 to 7. During the holiday, the coal-fired boilers in factories and power plants stopped working, which would significantly reduce $PM_{2.5}$ emissions from coal combustion. The much lower mass contribution of coal combustion in early October (as shown in Figure S9) coincided with this inference.

Revisions were made as follows.

In addition, we noted that the contribution of coal combustion was much lower in non-episode 2 than that in non-episode1 ($p<0.05$). The explanation is provided in Section 2 of the Supplement.

For details, please refer to lines 460-462, page 20.

The source apportionment results indicated that contribution of coal combustion in non-episode 2 was significantly lower than that in non-episode 1 ($p<0.05$). Bearing in mind the uncertainties caused by the lack of WSIs in summer, the lower contribution of coal combustion in non-episode 2 might be attributable to the National Day holiday from October 1 to 7. During the holiday, the coal-fired boilers in factories and power plants stopped working, which would significantly reduce $PM_{2.5}$ emissions from coal combustion. The much lower mass contribution of coal combustion in early October (as shown in Figure S10) coincided with this inference.

For details, please refer to Section 2 of the Supplement.